# Pulsed stimulated Brillouin microscopy enables high-sensitivity mechanical imaging of live and fragile biological specimens

Fan Yang [1,10] ✉, Carlo Bevilacqua [1,2], Sebastian Hambura [1], Ana Neves [1,11], Anusha Gopalan[1,11], Koki Watanabe[1], Matt Govendir[1,3], Maria Bernabeu[3], Jan Ellenberg [1], Alba Diz-Muñoz[1], Simone Köhler [1], Georgia Rapti[4,5,12], Martin Jechlinger [1,6] & Robert Prevedel [1,4,5,7,8,9] ✉

Brillouin microscopy is an emerging optical elastography technique capable of assessing mechanical properties of biological samples in a three-dimensional, all-optical and noncontact fashion. The typically weak Brillouin scattering signal can be substantially enhanced via a stimulated Brillouin scattering (SBS) process; however, current implementations require high pump powers, which prohibit applications to photosensitive or live imaging of biological samples. Here we present a pulsed SBS scheme that takes advantage of the nonlinearity of the pump–probe interaction. In particular, we show that the required pump laser power can be decreased ~20-fold without affecting the signal levels or spectral precision. We demonstrate the low phototoxicity and high specificity of our pulsed SBS approach by imaging, with subcellular detail, sensitive single cells, zebrafish larvae, mouse embryos and adult *Caenorhabditis elegans*. Furthermore, our method permits observing the mechanics of organoids and *C. elegans* embryos over time, opening up further possibilities for the field of mechanobiology.

Over the past decades, progress has been made in the fields of biomechanics to understand the relationship between mechanical forces, material properties and biochemical signals to regulate cellular function and tissue organization[1–5]. However, existing biophysical techniques[5–7] are predominantly based on direct physical contact and involve direct application of external forces to the sample. Therefore, they are either limited to sample surfaces or lack the required three-dimensional (3D) cellular resolution[5].

Recently, a type of optical elastography, namely, Brillouin microscopy (BM), has emerged as a nondestructive, label- and contact-free technique that allows probing of mechanical properties in a noncontact, label-free and high-resolution fashion[8,9]. It is based on light

[1]Cell Biology and Biophysics Unit, European Molecular Biology Laboratory, Heidelberg, Germany. [2]Collaboration for joint PhD degree between EMBL and Heidelberg University, Faculty of Biosciences, Heidelberg, Germany. [3]European Molecular Biology Laboratory Barcelona, Barcelona, Spain. [4]Developmental Biology Unit, European Molecular Biology Laboratory, Heidelberg, Germany. [5]Interdisciplinary Center of Neurosciences, Heidelberg University, Heidelberg, Germany. [6]MOLIT Institute for Personalized Medicine gGmbH, Heilbronn, Germany. [7]Epigenetics and Neurobiology Unit, European Molecular Biology Laboratory, Rome, Italy. [8]Molecular Medicine Partnership Unit, European Molecular Biology Laboratory, Heidelberg, Germany. [9]German Center for Lung Research (DZL), Heidelberg, Germany. [10]Present address: Shanghai Institute of Optics and Fine Mechanics, Chinese Academy of Sciences, Shanghai, China. [11]Present address: Collaboration for joint PhD degree between EMBL and Heidelberg University, Faculty of Biosciences, Heidelberg, Germany. [12]Present address: Epigenetics and Neurobiology Unit, European Molecular Biology Laboratory, Rome, Italy. ✉e-mail: fan.yang@embl.de; prevedel@embl.de

scattering of visible or infrared monochromatic (laser) light from gigahertz-frequency longitudinal acoustic phonons that are characteristic of the mechanical components of the material. The resulting Brillouin spectrum, that is, the frequency shift ($\Omega_B$), and linewidth ($\Gamma_B$) of the inelastically scattered light then provides information on the longitudinal modulus of the (bio-)material, assuming the relation between the refractive index (RI) and density is known. Although both the RI and the density may vary with conditions, their ratio $\rho/n^2$ does not vary substantially in biological materials[10–12]. Therefore, the value of Brillouin frequency shift and linewidth are often reported as direct indicators of the mechanical properties. When coupled to a confocal microscope[8], BM can achieve diffraction-limited resolution in 3D, which has enabled a wide range of applications in biology, including studies of intracellular biomechanics in living cells[11], of ex vivo[13,14] and in vivo tissues[12,15,16] and their components[17] as well as biomaterials[18] and disease diagnosis[19,20]. The typical signal levels in spontaneous Brillouin scattering, however, are relatively weak, due to the low scattering cross-section (~$10^{-10}$ to $10^{-12}$), which has hampered the development of high-speed and/or high-sensitivity Brillouin microscopes.

Recently, stimulated Brillouin scattering (SBS) has been introduced and shown to enable high spectral resolution and thus high mechanical specificity, as well as practical acquisition times of biological samples in BM of biological samples[21,22]. The technique is based on coherently driving, and thus enhancing, the phonon population inside a sample at a given frequency via two interfering laser (pump) beams[23] and scanning a frequency-tunable 'probe' laser over the Brillouin spectrum. In contrast to spontaneous Brillouin scattering, SBS enables spectral measurements at a higher resolution that are free of elastic background contributions, and thus enabling high signal levels while maintaining high mechanical specificity. The limitations of SBS are that it is normally restricted to relatively transparent (that is, non-absorbing) samples thinner than ~100–200 μm that can be optically accessed from two opposing sides. Although frequency-domain and impulsive SBS were previously used for Brillouin imaging of tissue phantoms at high spectral resolution[24–26], only a recent demonstration has achieved the necessary high spatial resolution and shot-noise sensitivity adequate for imaging of semi-transparent organisms, such as *Caenorhabditis elegans*[22]. However, this implementation used continuous wave (CW) laser illumination, therefore not taking full advantage of the nonlinearity of the stimulated process. Moreover, the required high illumination dosages (~265 mW in ref. 22) limit their use for live imaging of sensitive samples over extended time periods, which probably prohibits more widespread applications and its uptake in the life sciences.

In this Article, we address this major drawback of current SBS realizations by introducing a pulsed pump–probe approach. Combined with balanced detection and diligent parameter optimization, this allows us to achieve shot-noise limited performance yielding comparable signal-to-noise ratio (SNR), spectral resolution, image quality and speed compared to recent SBS demonstrations[22], yet at a substantially (>10-fold) lower illumination dosage. We further show that, in contrast to full-CW operation, pulsed SBS does not lead to sample heating or phototoxic effects in cultured single cells or during two-dimensional (2D) live imaging of entire organoids and *C. elegans* embryos over extended fields of view (up to $215 \times 215 \times 10$ μm$^3$) and time periods (up to 3 h). The high spatial ($0.57 \times 0.55 \times 2.58$ μm$^3$) as well as spectral (151 MHz) resolution and thus mechanical specificity further allowed us to distinguish up to three mechanically separate components in the focal volume in the zebrafish notochord and spinal chord region and to capture viscoelastic changes during organoid development.

## Results

### Pulsed SBS approach, setup and validation

The pulsed SBS approach is based on two CW, narrowband (100 kHz), yet tunable, amplified 780 nm diode lasers that illuminate the sample from two opposing sides through high (0.7) numerical aperture

(NA) objective lenses (Fig. 1a–d). The resulting interference pattern is determined by the spatial overlap of the pump and probe beams, which in our case are confined to $0.57 \times 0.55 \times 2.58$ μm$^3$ (Fig. 1d and Extended Data Fig. 1).

To illustrate the advantage of our pulsed SBS approach, consider the CW pump and probe power to have equal average optical power $P_{cw}$, while the pulsed pump and probe both feature a peak power of $P_{pulse}$, but a comparable average power of $P_{cw}$ (Fig. 1b). Here the SBS signal is proportional to $P_{cw} \times P_{cw} \times T_{period} = 1$ a.u., where $T_{period}$ is the pump–probe interaction time of the CW scheme. In contrast, for the pulsed scheme it is proportional to $P_{pulse} \times P_{pulse} \times T_{pulse} = \frac{T_{period}}{T_{pulse}} = \frac{1}{\text{duty cycle}}$ a.u., with $T_{pulse}$ being the pulse width. Therefore, the pulsed scheme yields an effective enhancement factor $E$ that scales inversely with the duty cycle (dc), or $E \approx 1/\text{dc}$ compared to the CW scheme. As the noise depends on the average probe power (which does not change), the pulsed scheme has $E$ times higher SNR at the same laser powers. This enhancement can be employed to decrease the pump power while keeping the same SNR and probe power.

Pulsing of both the pump and probe lasers is achieved via acousto-optic modulators (AOMs), which can be operated down to 40 ns width and 1.1 MHz repetition rate. In our work, we chose between 40 and 909 ns, which corresponds to a duty cycle of 4.4–100% or an averaged pump power dosage of 13–295 mW, respectively, and to a maximal $E \approx 22.7$ (Supplementary Note 1). For example, a pulsed scheme with 40 ns width for both pump (13 mW average power) and probe (5 mW average power) at 1.1 MHz repetition rate has the same SNR as a CW-based scheme with a 295 mW pump and a 5 mW probe (Fig. 1e,f). To record a Brillouin spectrum, the probe laser's frequency is rapidly tuned across the Brillouin resonance of the medium, and the scattering signal is detected as an increase, or gain, of the probe's intensity ($I_2$). The typical spectrum acquisition time is as low as 20 ms in biological samples. The detection unit comprises two photoreceivers for detection and reference, respectively, and a differential-input lock-in amplifier (LIA), whose operation parameters were carefully optimized (Supplementary Note 2 and Extended Data Fig. 2). The sample is then raster-scanned through the microscope's focus to obtain cross-sectional images of Brillouin shift ($\Omega_B$), linewidth ($\Gamma_B$) and gain ($G_B$).

First, we validated the pulse enhancement scheme by measuring the Brillouin water spectra and the SNR at different pulse length (hence different average power and duty cycle) and confirmed that SNR remains unchanged (Fig. 1e,f). Furthermore, the precision of the Brillouin shift and Brillouin linewidth measurements as a function of the acquisition time of the stimulated Brillouin gain (SBG) spectrum of water shows a slope close to −0.5, suggesting shot-noise limited performance within the measured range. We note this is on par with recent SBS demonstration[22] (Fig. 1g), but with ten times less total optical power. The pulsed SBS microscope also shows high spectral resolution (151 MHz, Methods and Extended Data Fig. 3) as well as close to shot-noise limited performance across a wide range of LIA dwell times (Supplementary Note 2). This demonstrates the pulse enhancement of our SBS approach and suggests state-of-the-art performance at effectively 22-fold reduced illumination powers.

### Cellular imaging with pulsed SBS

To translate the advancements of the pulsed-SBS scheme to biomechanical imaging, we first acquired Brillouin maps of NIH/3T3 mouse fibroblasts, primary human brain microvascular endothelial cells (HBMECs) and mouse embryonic stem (mES) cells, and compared them to images obtained when using quasi-CW illumination. Brillouin shift maps acquired with pulsed SBS (27 mW total power) showed overall high image quality and good SNR level (for example, on average 38 in the first z-plane of Fig. 2a), from which subcellular compartments with subtle mechanical differences such as the nucleoli could be discerned (Fig. 2a and Extended Data Fig. 4). From the spectral data, spatial maps

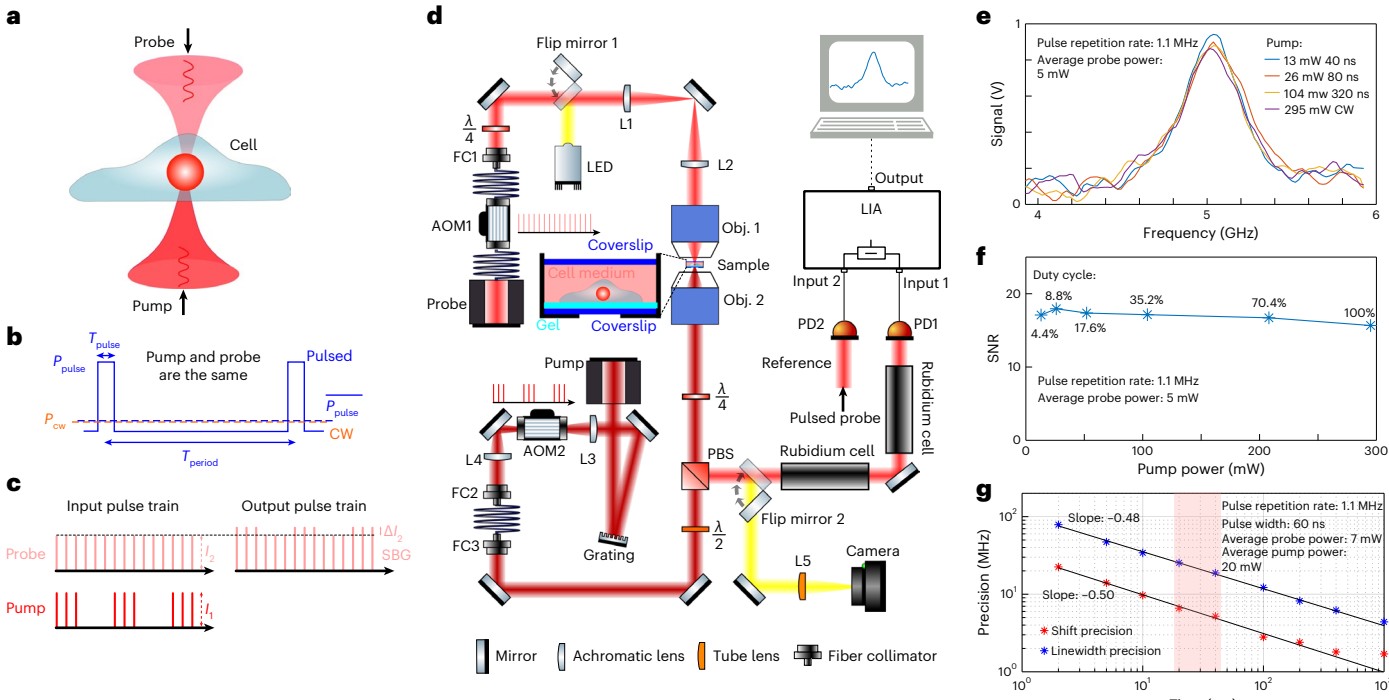

**Fig. 1 | Pulsed-SBS approach and performance. a**, Pump and probe beams with a slightly different frequency counter-propagate and are focused inside a biological sample. **b**, Schematic of the optical power against time for the pulsed scheme (blue solid line) and the CW scheme (orange dashed line). **c**, SBG detection scheme. The pump beam is modulated at high frequency (320 kHz) at which the resulting amplitude modulation of the probe beam due to SBG can be measured. **d**, Schematic of the SBS setup. The pump and probe beams are focused in the same position using high NA (0.7) objective lenses. The intensity of the probe beam is measured by a photodiode, connected to the LIA. Brightfield imaging can be performed by flipping two mirrors into the optical path (yellow). Widefield fluorescence image is obtained by adding an excitation filter after the LED and an emission filter before the camera (not shown). **e**, Pulsed-SBS spectra of water under different pulse width but at the same peak power (thus different average power) while keeping the same average probe power to 5 mW. The pump pulse width ranges from 40 ns to CW and the corresponding average power ranges from 13 to 295 mW. Integration time is 20 ms. **f**, Quantification of SNR as a function of duty cycle and average pump power. **g**, Precision of the Brillouin shift and linewidth as a function of the integration time of the SBG spectrum of water. The precision is calculated as the standard deviation of the Brillouin shift and linewidth determined from the Lorentzian fits of $n = 300$ SBG spectra measured sequentially. The shaded region marks the integration time used in our experiments.

of the Brillouin shift, gain and linewidth (Fig. 2e,g,h) both lateral as well as axial (Fig. 2f,i) can be plotted, demonstrating the high overall image quality and SNR. Staining with an established viability marker (pro-pidium iodide, PI) revealed membrane and/or chromosomal damage was absent in a pulsed-SBS imaged cell 15 h after imaging (Fig. 2a, $n = 5$, Extended Data Fig. 4d). Next, we sequentially imaged two fibroblast cells with pulsed and CW schemes. No cell blebbing and membrane damage was observed in the pulsed scheme (Fig. 2b). In contrast, while a Brillouin map acquired with SBS CW illumination (250 mW total power) gave similar contrast and quality, it showed direct effects of potential photodamage. Firstly, clear cell blebbing and membrane damage were observed in the CW scheme after imaging of five $z$-planes (Fig. 2c and Extended Data Fig. 4g). Secondly, the overall Brillouin shift ($\Omega_{\mathrm{B}}$) was elevated in different cell regions (nucleolus, nucleoplasm and cytoplasm). This trend was observed in both fibroblast ($n = 2$) and primary HBMECs ($n = 2$), which suggests sample heating (Fig. 2j). Fur-thermore, mES cells showed increased motility, which is indicative of cellular avoidance behavior (Extended Data Fig. 4h). Moreover, particle trapping and hence higher Brillouin shift in the cell medium is more evident in the CW scheme compared to pulsed SBS. Together, these experiments emphasize that in SBS-based imaging of single cells total illumination levels have to be carefully kept at low powers (<30 mW) to ensure cell viability.

## High-specificity SBS imaging in live zebrafish and *C. elegans*
Next, we set out to explore the high spectral resolution of our pulsed SBS for distinguishing different mechanical constituents in heterogeneous

living tissues. For this we acquired low-power, pulsed-SBS images in the tail region of live zebrafish larvae 3 days post-fertilization (3 dpf). Here the region surrounding the notochord is known to encompass an ultrathin, ~500-nm-wide, extracellular matrix (ECM) layer of high mechanical rigidity that supports the tissue structure[16]. The pulsed-SBS images revealed several tissue types including the ECM, the notochord and the surrounding muscle segments including the spinal chord region (Fig. 3c,d). Owing to the high spectral resolution of SBS, an asymmet-ric spectrum indicative of several peaks becomes evident in several tissue regions. Here the high spectral resolution of SBS reflects a high mechanical specificity[22] in the assumption that the ratio between RI and density does not substantially vary within cells and tissues, as previ-ously shown in zebrafish[12].

To aid in the efficient experimental evaluation and data analysis of the complex, multipeak Brillouin spectra, we developed a custom analysis approach and graphics processing unit (GPU)-enhanced fit-ting pipeline with a convenient graphical interface and substantially improved speed, which enabled real-time analysis and optimization of experimental parameters (Extended Data Fig. 5). With this we unam-biguously distinguish between single and multiple peaks in our spectra (Fig. 3e). Using our optimized tools, we observe a triple peak distribu-tion (Fig. 3f) of Brillouin shifts when analyzing the ECM region. We interpret these as an ECM-surrounding tissue component (5.34 GHz, L1), and a first (5.63 GHz, L2) and second (6.63 GHz, L3) acoustic mode that we attribute to the ECM. We hypothesize the L2 and L3 components to represent hybrid acoustic modes which involve both longitudinal and transverse motions, due to the ECM being thinner than the optical

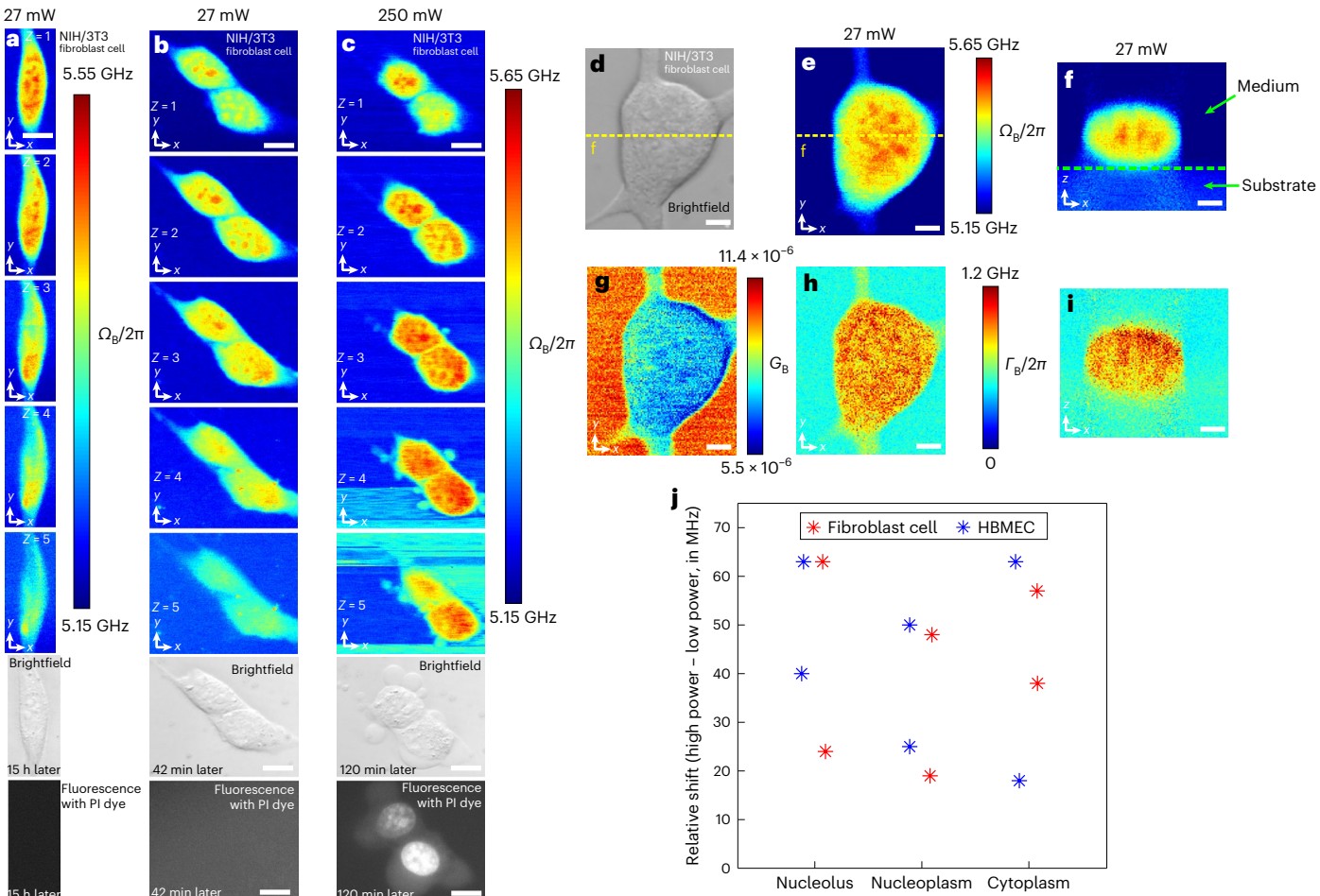

**Fig. 2 | Pulsed-SBS imaging of cultured cells. a**, Three-dimensional sections of Brillouin shift images of a NIH/3T3 fibroblast cell under 20 mW pump and 7 mW probe with a z-step of 1 μm. The coregistered brightfield image and fluorescence image (15 h after Brillouin imaging) of the cell stained with PI dye are shown below. **b**, Three-dimensional Brillouin shift images of two fibroblast cells under 20 mW pump power and 7 mW probe power with a z-step of 1 μm. The brightfield and fluorescence images (42 min after Brillouin imaging) are shown below the Brillouin images. **c**, Three-dimensional Brillouin shift images of the same two cells under 240 mW pump power and 7 mW probe power after **b** with a z-step of 1 μm. Scale bars in **a**–**c** are 10 μm. Note that the PI dye, a viability marker that indicates damaged membranes, is staining the nucleus. **d**, Brightfield image of

a NIH/3T3 fibroblast cell. **e,g,h**, The Brillouin shift ($\Omega_B$) (**e**), Brillouin gain ($G_B$) (**g**) and Brillouin linewidth ($\Gamma_B$) (**h**) images of the x–y plane. **f,i**, The Brillouin shift (**f**) and Brillouin linewidth (**i**) images of the x–z plane along the dashed yellow line in **d**. The dashed green line in **f** marks the substrate (that is, PAA-based gel) boundary. Scale bars in **d**–**i** are 5 μm. **j**, Relative frequency up-shift of CW SBS with respect to pulsed SBS of three regions (nucleolus, nucleoplasm and cytoplasm) for ($n = 4$) cells between 27 mW power and 250 mW power. The average up-shifts in the nucleolus, nucleoplasm and cytoplasm regions are 47.5 MHz, 35.5 MHz and 44.0 MHz, respectively. All image pixel steps and pixel time are 0.25 μm × 0.25 μm and 20 ms, respectively.

wavelength, since such hybrid acoustic mode-induced splitting is often observed in subwavelength optical waveguides[27]. Note that previous work[16] in the zebrafish notochord was only able to distinguish the higher-shift, and thus well-separated, second ECM mode. Pulsed SBS is therefore capable of identifying three distinct and specific biomechanical components inside a diffraction-limited focal volume and inside a living specimen. We also observed double-peaked Brillouin spectra when analyzing the central canal regions in the spinal chord (Fig. 3g), which we interpret as cerebrospinal fluid (5.03 GHz, L1) and presumable beating cilia[28]. We further highlight that here we are able to measure both components inside the point-spread function (PSF) simultaneously, that is, for every image pixel, and during active cerebrospinal fluid flow.

We further validated the quality of our pulsed-SBS microscopy by imaging live, anesthetized wild-type *C. elegans* at the young adult stage. As evident from close-up images of the pharyngeal region (Fig. 3i,j), pulsed-SBS with 27 mW total power yields similar SNR and overall contrast compared to standard CW scheme at ten-fold reduced

illumination power. Here, under CW illumination, optical trapping of dust in the illumination laser light focus, particularly in the media and lumen, were more common and the specimens imaged showed enhanced motility behavior despite anesthesia, probably a stress response in the animal[29]. Next, we used our pulsed SBS to observe spatial differences in Brillouin shift in the gonad region of the nematode at high resolution (Fig. 3k). The Brillouin shift of meiotic nuclei decreased during oocyte maturation. While a thin shell of lower Brillouin shift nucleoplasm was wrapped around a higher Brillouin shift nucleolus in early oocyte nuclei, both nucleoplasm and nucleolus regions were with lower Brillouin shift in late oocytes. This decrease in Brillouin shift correlates with an increase in chromatin compaction from early to late oocytes.

To further highlight the advantage of low-power pulsed SBS, we imaged live mouse embryos in the zygote and eight-cell stage (Extended Data Fig. 6). We also obtained high image quality and subcellular details with pulsed SBS, while the high optical power of CW SBS trapped the embryo and thus prohibited Brillouin imaging.

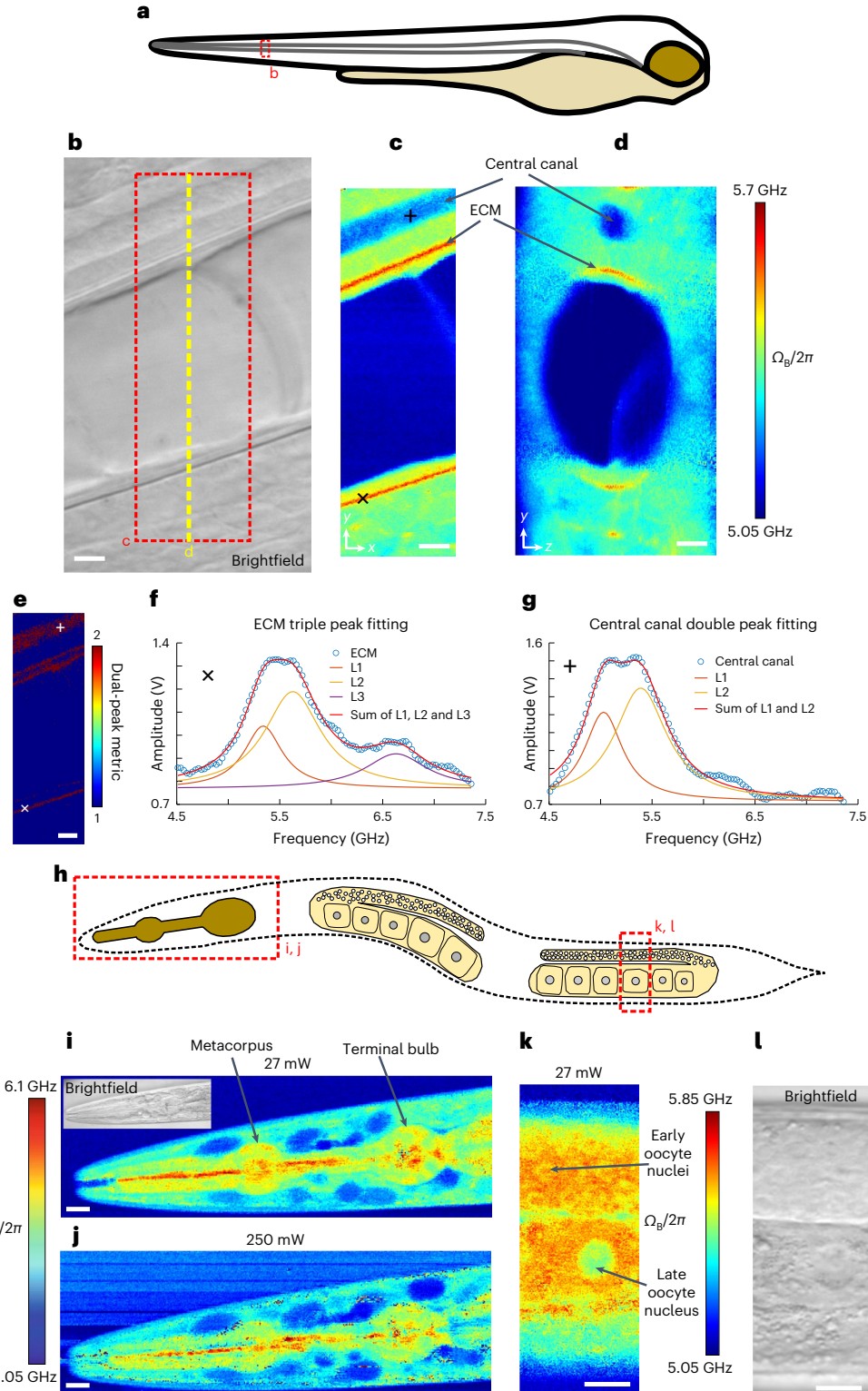

**Fig. 3 | Pulsed-SBS imaging of zebrafish larvae and young adult *C. elegans*.**
**a**, Illustration of a zebrafish larva, indicating the imaging region (dashed red box).
**b**, Brightfield image of the notochord region of a zebrafish larva at 3 dpf stage.
**c**, Brillouin shift image of the *x*–*y* plane of the region marked by a red dashed box in
**b**. **d**, Cross-sectional Brillouin shift image in the *y*–*z* plane along the yellow dashed
line in **b**. Scale bars in **b**–**e** are 5 μm. In **c** and **d**, the pixel steps in *x* and *y* directions
are both 0.2 μm and the pixel time is 40 ms. **e**, Multi-peak metric map of the
region marked by a red dashed box in **b**. **f**, A representative Brillouin spectrum
in the high-shift region, indicative of ECM, marked by a cross in **c** and **e**. The raw
spectrum and its triple-peak fitting are shown (L1–3). Note that an arbitrary offset
was added to L1–3 for visualization. **g**, A representative Brillouin spectrum inside

the central canal in the spinal chord region marked by a plus sign in **c** and **e**. The raw
spectrum and its double-peak fitting (L1 and L2) are shown. Note that the offset
was added to L1 and L2 for visualization. **h**, Schematics of a young adult *C. elegans*.
The head and gonad regions imaged are marked by dashed red boxes. **i,j**, Brillouin
shift images of the head region under 27 mW power (**i**) and 250 mW power (**j**). Note
that slight movements of the nematode during high-power imaging blur out the
contrast (c.f. **i**). In **i** and **j**, the pixel steps in *x* and *y* directions are both 0.5 μm and
the pixel time is 20 ms. **k,l**, Brillouin shift image (**k**) and coregistered brightfield
image (**l**) of the gonad region marked by a dashed red box in **h**, showing oocyte
nuclei from different meiotic stages (arrows). In **k**, the pixel steps in *x* and *y* directions
are both 0.25 μm and the pixel time is 20 ms. Scale bars in **i**–**l** are 10 μm.

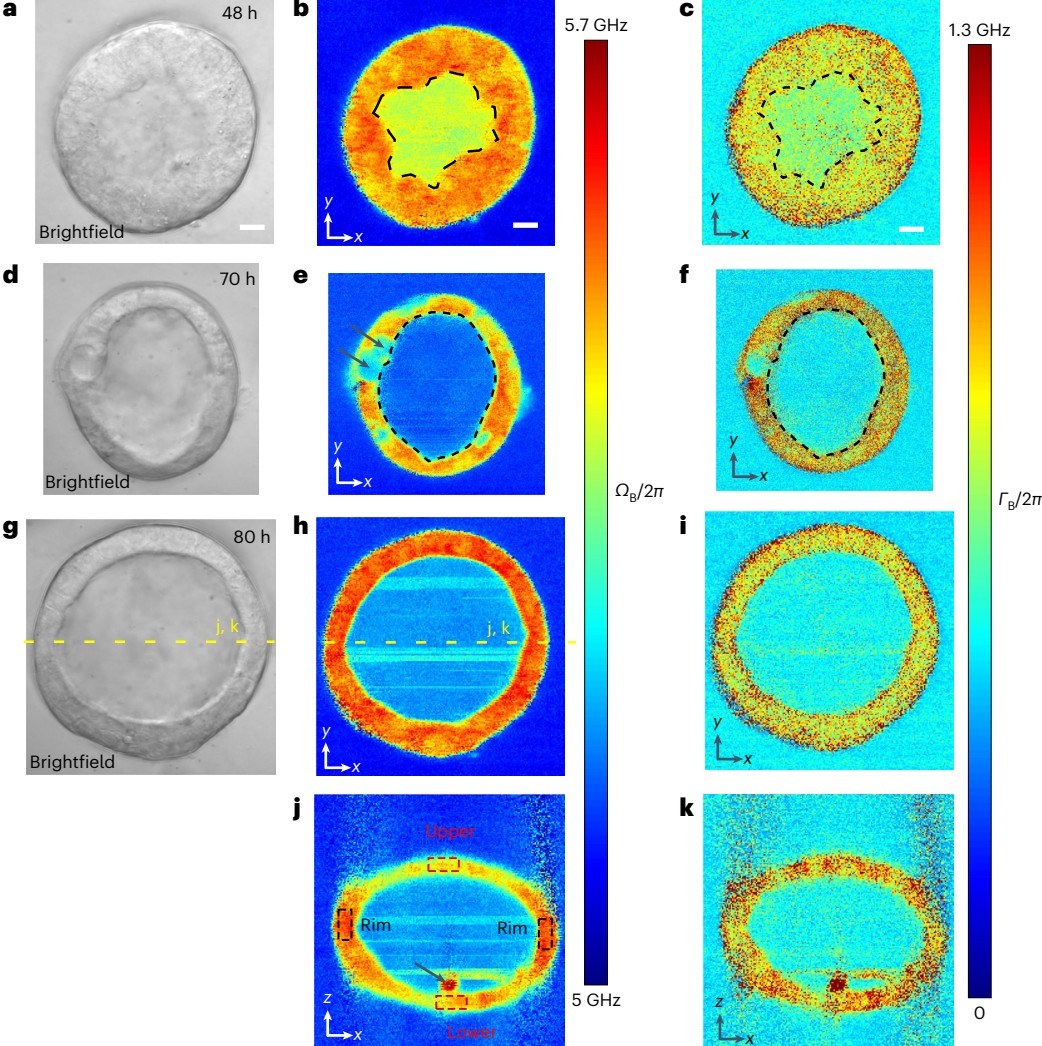

**Fig. 4 | Pulsed-SBS imaging of mouse mammary gland organoids.**
**a**, Brightfield image of an organoid 48 h post seeding singularized mammary epithelial cells into basement membrane matrix. **b,c**, Brillouin shift image (**b**) and Brillouin linewidth (**c**) image of the organoid in **a**. In **b** and **c**, the pixel steps in $x$ and $y$ directions are both 0.5 μm and the pixel time is 20 ms. **d**, Brightfield image of an organoid 70 h post seeding. **e,f**, Brillouin shift image (**e**) and Brillouin linewidth (**f**) image of the organoid in **d**. The average Brillouin linewidth in the lumen in **f** (FWHM 488 MHz) is 200 MHz decreased compared to **c** (FWHM 688 MHz) (representative of $n = 3$ organoids). Arrows in **e** highlight a dividing cell in cytokinesis. Dashed lines in **b–f** demarcate organoid lumen. In **e** and **f**, the pixel steps in $x$ and $y$ directions are both 0.25 μm and the pixel time is 20 ms.

**g**, Brightfield image of an organoid 80 h post seeding. **h,i**, Brillouin shift image (**h**) and Brillouin linewidth (**i**) image of the organoid in **g**. **j,k**, Brillouin shift image (**j**) and Brillouin linewidth (**k**) image of the $x–z$ plane marked with a dashed yellow line in **g** and **h**. The average Brillouin shift of the bended rim regions (black dashed boxes in **j**, average 5.56 GHz) shows a ~100 MHz up-shift compared to the average shift of the upper and lower cap of the epithelium (red dashed boxes in **j**, average 5.46 GHz) (representative of $n = 4$ organoids). A piece of debris, potentially a dead, extruded cell is marked with an arrow in **j**. In **h–k**, the pixel steps in $x$ and $y$ directions are 0.5 μm and the pixel time 20 ms. Scale bars in all the images are 10 μm.

## Viscoelastic imaging of mouse mammary gland organoids

To demonstrate the applicability of our method for medically relevant test systems, we acquired pulsed-SBS images of mouse mammary epithelial organoids (Fig. 4). The epithelial tissues and cells were well resolved in lateral as well as axial cross-sections. Quantifying the Brillouin linewidth ($\Gamma_B$) we observed a transient decrease in linewidth (~200 MHz) in the organoid lumen between early (48 h post-seeding the single cells into basement membrane matrix) (Fig. 4b,c) and later phases (70 h and 80 h post-seeding) (Fig. 4e–k) of development. In this organoid system, lumen formation is generated through cell division[30] and hollowing[31], which typically involves endocytosis of membrane vesicles to extend the apical membrane[30] as well as hydrostatic pressure, which builds through the accumulation of ion channels and pumps at the apical plasma membrane[32,33]. These processes may explain the observed increased Brillouin linewidth in the early phase of organoid

growth, when a nascent lumen is formed (Fig. 4b,c). We furthermore captured a putative cell division event during late cytokinesis (Fig. 4d). Remarkably, these cells showed a substantial decrease in Brillouin shift (arrows in Fig. 4e), in line with observations in the Madin-Darby canine kidney cell line at the division elongation/cytokinesis stage in 2D cell culture conditions as measured by traction-force microscopy[34].

Finally, in 80-h-old, established organoids, we observed differential Brillouin shift in axial cross-sectional images, with the side epithelium displaying a 100 MHz higher shift (Fig. 4j). Overall, organoids remained viable after pulsed-SBS imaging, as confirmed by brightfield microscopy acquired at 24 and 48 h after the imaging experiments, that showed intact organoids and no detectable blebbing or debris originating from dead cells within the lumen. Note that, even with only 27 mW power, particle trapping due to optical tweezer effect leads to stripe-like artifacts in the lumen of the organoids (Fig. 4h–j).

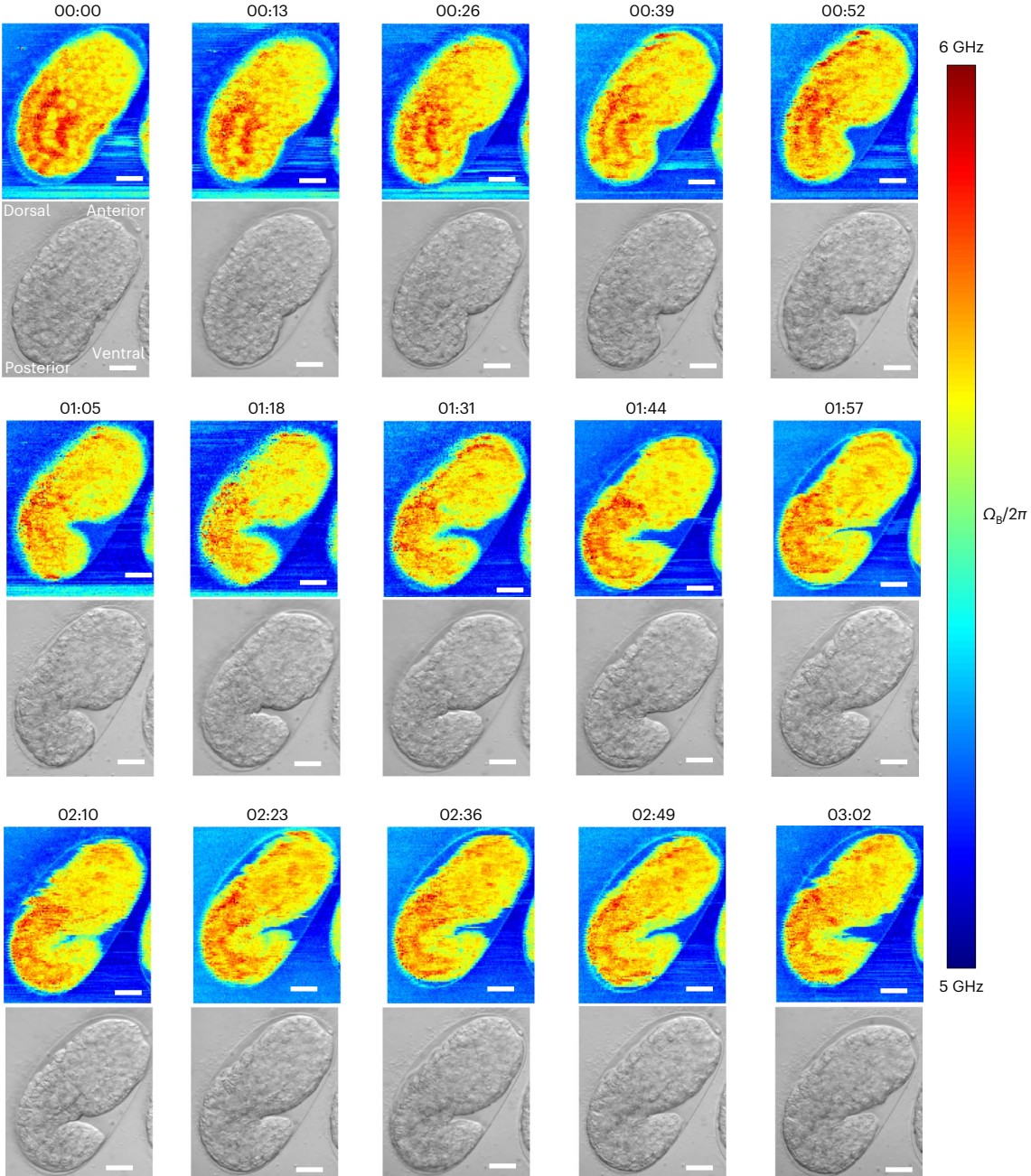

**Fig. 5 | Time-lapse pulsed-SBS imaging of *C. elegans* embryo development.** Brillouin shift (top) and brightfield (bottom) images over a 3-h time span and at 13-min time interval (representative of *n* = 3 embryos). The pixel steps in *x* and *y* directions are 0.5 μm and the pixel time is 20 ms (total time per image 288 s). Scale bars in all the images are 10 μm. The embryo at the start of the time lapse is at the late 'ball-stage', slightly before the bean stage, or approximately at 300 min post-fertilization (mpf). Imaging time lapses presented here conclude at ~480 mpf, when embryos start their first spontaneous muscle contractions, known as 'twitching'. Embryonic axes: top right is anterior, bottom left is posterior, right is ventral and left is dorsal.

## Pulsed SBS enables live imaging of *C. elegans* embryos

As a final demonstration of the low phototoxicity of our pulsed-SBS method and its capability to visualize dynamic tissue properties throughout morphogenesis, we captured longitudinal and 2D mechanical information of the developing *C. elegans* embryo, from ~300 min to 480 min post-fertilization (mpf), that is, from 'bean' stage to '1.5 fold' stage (Fig. 5 and Supplementary Video 1). This is the period of embryonic development when tissue morphogenesis starts, after the majority of embryonic cell divisions have concluded[35]. We acquired 2D Brillouin time-lapse images over a ~65 × 55 μm² field of view within ~4.8 min and at 13 min time intervals (that is, over 15 separate time points). Analyzing our Brillouin time-lapse data, we observed differential Brillouin shifts across embryonic body parts and embryonic stages. During *C. elegans* embryonic morphogenesis, higher Brillouin shifts mark the posterior part of the embryo, which mostly consists of endodermal cells that will form the intestine (Fig. 5)[35,36]. No photodamage or phototoxicity was observed at ~27 mW of average laser power as confirmed by brightfield images of the embryos post-acquisition (*n* = 3 embryos). *C. elegans* embryos remained active and viable after pulsed-SBS imaging. All imaged embryos undergo twitching and normal development past the stage of '4-fold' and eventually hatched into viable larvae. This represents a substantial improvement over standard CW SBS employing ~265 mW, where embryo damage and death was observed following the acquisition of a few 2D images (*n* = 3 embryos, Extended Data Fig. 7).

## Discussion

To summarize, here we presented a scheme for stimulated BM that fully exploits the nonlinearity of the pump–probe interaction. In our work we harnessed the improved efficiency of our approach along with diligent optimization of the signal detection to substantially reduce the required total illumination power and thus enable imaging of a wide range of photosensitive biological samples over extended time periods that could otherwise be damaged by the high laser powers required by previous SBS implementations.

Our pulsed SBS approach achieves ~20-fold signal enhancement at a noise level only 2-fold higher than the shot noise and compared to state-of-art CW implementation[22]. Therefore, ten-fold lower power illumination could be employed. Ensuring low photoburden is of utmost importance when establishing imaging and spectroscopy methods for applications in biomedicine. Here we note that our ten-fold/20 dB power reduction compared to the shot noise limited CW scheme is substantially higher than recently achieved quantum-enhanced spectroscopy schemes for Raman[37] (35% SNR enhancement) and SBS[38] (3.5 dB SNR enhancement).

While the spectral and thus overall image acquisition time of pulsed SBS (~20 ms) is notably slower than recently developed line-scan BM[39–41], we note that it is still faster than standard confocal implementations[11,12,42,43] yet yields improved spectral resolution and thus linewidth fidelity, and in principle allows information to be obtained on the mass density[22]. Therefore, by combining our pulsed-SBS microscopy with methods that measure the RI in 3D such as optical diffraction tomography[44,45], the complex longitudinal modulus can be quantitatively obtained. We note that the high-frequency longitudinal modulus measured by Brillouin scattering is, however, fundamentally different from the low-frequency, Young's modulus often used in mechanobiology[9].

Future hardware developments will allow full exploitation of the possible enhancement of our pulsed-SBS approach, which is currently limited by the input power of the fiber-pigtailed probe AOM (Supplementary Note 1). In particular, nanosecond-pulsed laser light generated by faster electro-optic modulator, together with smaller repetition rate (for example 100 kHz), could unlock the remaining untapped potential. Here we envision that diligent illumination scheme design could achieve overall enhancements of up to ~10.000-fold, which could be translated to practical spectral acquisition times of as low as ~20 µs at <1 mW average illumination power only (Supplementary Note 1). Such improvements could open new avenues and further applications for studying biomechanics in 3D with high spatio-temporal resolution and mechanical specificity in living biological specimen.

## Online content

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

## Methods

Animal work in this research was carried out at the European Molecular Biology Laboratory (EMBL). All animal care and procedures performed in this study conformed to the EMBL Guidelines for the Use of Animals in Experiments and were reviewed and approved by the Institutional Animal Care and Use Committee.

### SBS theory

Instead of employing the interaction between photons and phonons generated by thermal agitation in spontaneous Brillouin scattering, SBS uses two light beams. Pump ($\omega_1$) and probe ($\omega_2$) beams with a slightly different frequency counter-propagate and are focused in the sample, sharing the same focal volume. The beating between the pump and probe generates an interference fringe pattern, whose contrast is time-varying at the frequency given by the difference between the pump and probe frequency. Due to electrostriction, the interference fringe modulates the density of the sample and hence generates an acoustic wave. Because of the photoelasticity effect, the acoustic wave changes the RI of the sample and generates a moving Bragg grating that can transfer the energy from the pump to the probe beam. Therefore, this results in an SBG for the probe beam, while for the pump this represents a stimulated Brillouin loss (SBL) process. The process turns resonant when the beat pattern moves at the sound velocity in the sample, which is realized for a well-defined frequency difference ($\Omega_B = \omega_1 - \omega_2$) between the pump and probe beams.

The SBG or SBL spectrum is given by $G(\Omega) = \pm\eta \times g(\Omega) \times l \times I_1$, where $\eta$ is the overlap efficiency of the pump and probe beams in the sample, $\pm g(\Omega)$ is the SBS gain or loss factor well described by a Lorentzian function, $l$ is the interaction length of the two counter-propagating laser beams in the sampled volume, and $I_1$ is the intensity of the pump beam at $\omega_1$. As for the spontaneous Brillouin scattering spectrum, the close relationship between the SBG or SBL spectrum and the complex longitudinal modulus of the probed volume allows one to locally extract the high-frequency viscoelastic response of the medium. This relationship is described by $M^* = \rho \times (\lambda_1/2n)^2 \times \Omega_B^2 \times (1 + i\Gamma_B/\Omega_B)$, where $M^*$ is the complex longitudinal modulus, $\lambda_1$ is the wavelength of the beam at $\omega_1$, and $n$ and $\rho$ are the RI and the mass density of the medium, respectively. Similar to other nonlinear optical techniques, SBS inherently provides optical sectioning in three dimensions owing to the nonlinearity of SBS in the total irradiance intensity.

### Pulsed SBS imaging setup

A detailed schematic of our pulsed-SBS microscope is shown in Fig. 1d. Two tapered-amplified CW external-cavity diode lasers (TA-Pro, Toptica) with linewidth of 100 kHz are used as pump and probe beams. The wavelength of the pump laser is kept at 780.24 nm, which is at the D2, $F = 3$ absorption line of Rb[85]. To get the SBG spectrum, the probe laser frequency is scanned across the Brillouin gain peak ($\Omega_B$) by sawtooth modulation of the laser piezo. The probe beam ($s$-polarization) is amplitude modulated by a fiber-pigtailed AOM1 (#TEM-250-780-2FP-HP, Brimrose) to generate a pulse train (for example, 60 ns pulse width, 1.1 MHz repetition rate). An aspheric lens (#49-115, Edmund) is used to collimate the output of the AOM to a collimated beam with a diameter of 6.7 mm ($1/e^2$, theoretical). The probe beam is then right circularly polarized by a quarter wave plate. Two mirrors, conjugated to the back and the front focal plane of the probe objective (Obj. 1) by means of two achromatic lenses (L1 and L2: #49-356, Edmund), respectively, allow for fine position and angle adjustment of the beam inside the sample to aid in the alignment between the pump and probe overlap.

For the pump beam, a holographic narrowband reflective Bragg grating (SPC-780, OptiGrate) is employed to clean up the pump laser's amplified spontaneous emission noise. The filtered pump beam with $s$-polarization is focused into a free-space AOM2 by an achromatic lens (AC254-100-B-ML, Thorlabs) and re-collimated by another achromatic lens (AC254-100-B-ML, Thorlabs). AOM2 is not only used for pulse

generation (for example, 60 ns pulse width, 1.1 MHz repetition rate) but also for envelope amplitude modulation at 320 kHz frequency so that the pump modulation can be transferred to the probe by SBG and be measured by an LIA (MFLI 5 MHz, Zurich Instruments). A single-pole, single-throw analog switch (#NC7WB66K8X, Onsemi) is used to electronically combine the pulse and the envelope amplitude signals. The pulse signal (with 60 ns pulse width, 1.1 MHz repetition rate) is used as a digital enable input that allows the input signal (sinusoidal envelope modulation at 320 kHz) to pass (or not). The pulsed pump beam is coupled into a fiber by a fiber coupler FC2 (FiberDock, Toptica) for beam delivery and is then collimated with a diameter of 6.7 mm ($1/e^2$, theoretical) by a fiber collimator (#49-115, Edmund). Next, the beam is $p$-polarized by a half wave plate and transmitted through a polarizing beam splitter (CCM1-PBS25-780M, Thorlabs). The polarization of the beam is then changed to left circular polarization by a quarter wave plate.

The right-circular polarized probe beam and left-circular polarized pump beam counter-propagate and are focused into a sample at the same position by two 0.7 NA objective lenses (LUCPLFLN60X, Olympus). Note that Obj. 2 is mounted on a 3D translational stage for optimization of the overlap between pump and probe beams. Three-dimensional images are obtained by raster-scanning the 3D piezo stage (L3S-D10300-XY300Z300, nanoFaktur) onto which the sample is mounted. After interacting with the pulsed pump beam, the SBG probe beam is collected by Obj. 2 and its polarization is changed from right-circular polarization to $s$-polarization by the quarter wave plate. The $s$-polarization SBG probe beam is then reflected by the polarization beam splitter and passes through two rubidium cells (SC-RB85-(25×150-Q)-AR, Photonics Technologies) before being detected by a photodiode PD1 (FDS1010, Thorlabs). The output of PD1 is split into direct current (DC) and alternating current (AC) components by a bias-tee (ZFBT-4R2GW+, MiniCircuits). The DC component is measured by a data acquisition (DAQ) card (National Instruments) for gain calculation and balance detection. The AC signal is filtered by a low-pass filter (LPF-B0R35+, MiniCircuits) and then measured by the differential input 1 of the LIA. A fiber coupler takes out 50% of the pulsed probe beam for balance detection. The other 50% go through a variable neutral density attenuator and is detected by PD2 (FDS1010, Thorlabs). The output of PD2 is also split into DC and AC components by a bias-tee (ZFBT-4R2GW+, MiniCircuits). The DC component is measured by the DAQ card as a reference beam for balance detection. The AC signal is filtered by a low-pass filter (LPF-B0R35+, MiniCircuits) and then measured by the differential input 2 of the LIA. It should be emphasized that the LIA is set to differential input mode for balance detection, which can substantially decrease the detection noise (Extended Data Fig. 2). The LIA demodulated signal is acquired by the DAQ card. The reflection of the pump beam and the stray pump light can be filtered out by two rubidium cells. The Brillouin imaging data were acquired by custom-written LabView code.

To acquire brightfield and widefield fluorescence images, flip mirror 1 is flipped down and flip mirror 2 is flipped up. A lamp (X-Cite 120Q, Excelitas) with a light guide and a coupling adapter (X-Cite Light Guides, Excelitas) is used for brightfield imaging and the adapter output is conjugated to the back focal plane of Obj. 1 for Köhler illumination. The brightfield image is recorded by a complementary metal-oxide-semiconductor (CMOS) camera (CM3-U3-31S4M, Flir).

To acquire widefield fluorescence images of the PI dye for cell viability tests, a green light-emitting diode (LED; M530L4, Thorlabs) is butt-coupled to a light guide with a coupling adapter (X-Cite Light Guides, Excelitas) and used as an illumination light source. An excitation bandpass filter (ET535/50m, Chroma) and detection long-pass filter (ET605LP, Chroma) are placed after the LED and before the tube lens, respectively, for fluorescence detection.

### Pulsed-SBS system characterization and image acquisition

**Optimization of pump–probe overlap.** To maximize the overlap efficiency $\eta$ of the pump and probe beam as well as to ensure the best

axial resolution, the correction collars of the two 0.7 NA objectives require fine-tuning. First, the overlap between the pump and probe beam is roughly optimized (that is, pump and probe are focused in the same position in the sample) by maximizing the coupling probe power out of FC2 in Fig. 1d. Next, the correction collar of Obj. 2 is adjusted to make the brightfield image of a specimen (for example, a cell) as sharp as possible. While doing so, the z position of the sample needs to be adjusted, iteratively, to keep it in focus. After this procedure the best aberration correction for Obj. 2 is achieved and Obj. 1 can be optimized by maximizing the overlap efficiency $\eta$ as described before.

**Spatial and spectral resolution measurements.** The lateral (that is, x and y) resolution is characterized by measuring a polydimethylsiloxane (PDMS) bead in 1% (w/v) agarose (Extended Data Fig. 1a–d). The Brillouin frequency shifts of the agarose and PDMS bead are 5.12 GHz and 4.07 GHz, respectively. We plot the fitting signal amplitude at 5.12 GHz in Extended Data Fig. 1b. Then we fit the amplitude across the edge of the bead in x and y directions with erf function. Therefore, we get the x and y full width at half maximum (FWHM) resolution of 0.57 μm and 0.55 μm, respectively.

Because of the large RI difference between PDMS bead (RI 1.39) and agarose (RI 1.33), the aberration is high in the region close to the axial boundary of PDMS bead and agarose. We characterize the axial resolution by measuring immersion oil sandwiched between two coverslips (Extended Data Fig. 1e,f). The Brillouin frequency shift of the oil is 7.04 GHz and the frequency shift of the cover glass is more than 15 GHz; thus, the cover glass Brillouin signal is outside our scanning range (5–9 GHz). We scan the focus point through the boundary of oil and glass, and the signal amplitude at 7.04 GHz is shown in Extended Data Fig. 1f. The z axial resolution is calculated to be 2.58 μm by fitting the erf function to the experimental data.

To measure and calibrate the spectral resolution of the pulsed-SBS microscope, double-distilled water SBG signal was recorded at 23 °C and a representative spectrum is shown in Extended Data Fig. 3. We note that, in our setup, the frequency counter (Keysight 53210 A) gate time (100 ms) is much longer than the experimental dwell time of each frequency step (that is, 200 μs for 20 ms pixel time with 100 frequency points); therefore, the frequency difference between the pump and probe cannot be determined in real time. Therefore, the start and end frequency of the scanning range was measured by the frequency counter at very slow scanning speed. Since there is an offset between slow and fast scanning, we take the Brillouin frequency shift of water under slow scanning as a reference. The water spectrum at fast scan speed shown in Extended Data Fig. 3 was calibrated by subtracting the differential frequency (that is, the Brillouin frequency shift between the fast and slow scanning) from the measured frequency. The water FWHM linewidth is 459.2 MHz for the integration time of 20 ms and LIA noise-equivalent bandwidth of 200 Hz. Therefore, the spectral resolution of the microscope is calculated to be 151 MHz by subtracting our measured water FWHM to the low-NA water FWHM (308 MHz measured in ref. 22). Note that the water FWHM linewidth is measured to be 410 MHz (corresponding to 102 MHz spectral resolution) when the integration time is 40 ms and LIA noise-equivalent bandwidth is 200 Hz, which is in a good agreement with the measured high (0.7) NA spectral broadening of the CW scheme[22]. This also implies that the additional 49 MHz frequency broadening is due to the averaging effects of the LIA. Furthermore, we numerically evaluated the Brillouin shift and width accuracy under different spectral scan ranges and confirmed that scan ranges of greater than ~2 GHz do not lead to improved fidelity (Extended Data Fig. 8).

**Determination of signal SNR.** In our pulsed-SBS microscope, there are three types of contrast: Brillouin frequency shift $\Omega_B$, Brillouin linewidth $\Gamma_B$ and Brillouin gain $G_B$. $\Omega_B$ and $\Gamma_B$ of each pixel are respectively the peak frequency and FWHM linewidth of the Lorentzian fitting

of the measured spectrum. $G_B$ of each pixel is the ratio between the amplitude of the Lorentzian fitting and the DC component of the photodiode output.

The signal intensity is the amplitude of the Lorentzian fitting from the measured spectrum. The noise is the standard deviation of the measured spectrum when the pump–probe frequency difference is tuned away from the Brillouin frequency shift. For example, to get the water SBG spectrum, the frequency difference was scanned from 4 GHz to 6 GHz. The signal intensity is the amplitude of the Lorentzian fitting. The noise is the standard deviation of the spectrum outside the Brillouin interaction (that is, 7–9 GHz).

### SBS spectral data analysis

The data acquired from the microscope are stored as a unidimensional array. The data are reshaped into a 2D array of dimension [frequency;pixels], and a high-pass filter is applied (cutoff frequency 50 Hz, corresponding to the typical acquisition time of a single spectrum). The recorded data are then processed using a customized version of Gpufit[46] to which a Lorentzian function and a sum of two Lorentzian functions were added as fitting functions. Each Lorentzian fit gives access to the shift, width and amplitude of the peak. To plot an image or an image stack from these data, the array is reshaped, every second line is shifted (by a number of pixels equivalent to ~100 ms) to compensate for the delay between desired and actual stage position and every other plane is flipped (z-scan pattern).

Except for the modified version of Gpufit (v1.2.0), which is written in C++ and Cuda 11.5, the rest of the processing is done in MATLAB 2021b. This allowed the development of a graphical user interface to explore the acquired data, to detect problems in the acquisition and to verify the goodness of the fits. This graphical user interface displays two images made from a selected parameter (shift, width, amplitude, error and so on), and clicking on a pixel in these images displays the underlying acquired spectra and the fitted function (Extended Data Fig. 5). In some cases, it is possible that the recorded spectra is a 'double peak', that is, the sum of two Lorentzian functions. In that case, Gpufit is used to directly fit the sum of two Lorentzian functions. To distinguish potentially asymmetrical peaks from true double peaks, we compute the derivative of the fitted sum of Lorentzian functions, and evaluate whether it intersects more than once with the $y = 0$ line, in which case it is indicative of at least two local extrema. Furthermore, we slightly shift the derivative offset and repeat the test. In case we also get intersections with $y = 0$, it points to one local extrema in the spectrum and a 'bump' on its side. Both these situations are clearly impossible to have with only one, either symmetrical or asymmetrical, peak. We found that shifting the derivative offset from −0.5 to +0.5 allowed us to classify 99.9% of pure water spectra as a single peak.

On our workstation (11th Gen Intel(R) Core(TM) i9-11900K @ 3.50 GHz and Nvidia GeForce GTX 1050 Ti), using the GPU and Gpufit instead of using the default MATLAB fitting function enables a speed-up of the total processing time of ~340 times (Supplementary Table 1).

### Sample preparation and imaging parameters

**Cells (including staining and controls).** Mouse fibroblasts (NIH/3T3-CRL-1658 from ATCC) were grown in media composed of Dulbecco's modified Eagle medium (DMEM) with 4.5 g l⁻¹ glucose, 10% fetal bovine serum (Gibco, cat. no. 26140079) and 100 U ml⁻¹ penicillin–streptomycin (Gibco, cat. no. 15140122). They were seeded on polyacrylamide (PAA) gels with 12 kPa hydrogel stiffness (Matrigen, cat. no. SV3510-EC-12) at a density of 4,000 cells cm⁻² and allowed to adhere overnight before imaging.

Mouse embryonic stem (mES) cells (sox1–GFP–mES cells from Austin Smith's lab, University of Exeter) were maintained as undifferentiated stem cells in medium composed of high-glucose DMEM (Gibco, cat. no. 11960044) supplemented with 15% ES-qualified fetal bovine serum (Merck, cat. no. ES009B), 1× non-essential amino acids

(Gibco, cat. no. 11140035), 1 mM sodium pyruvate (Gibco, cat. no. 11360039), 0.1 mM β-mercaptoethanol (Gibco, cat. no. 21985023), 1 U ml⁻¹ leukemia inhibitory factor (from EMBL protein expression and purification core facility), 2 mM L-glutamine (Sigma, cat. no. G7513) and 100 U ml⁻¹ penicillin–streptomycin. Medium was refreshed every day, and the cells were subcultured every second day with a seeding density of 10,000 cells cm⁻². PAA gels were coated with 0.1% bovine skin gelatin type B (Sigma, cat. no. G9391) in phosphate-buffered saline (PBS) for 1 h at 37 °C before seeding mES cells at a density of 10,000 cells cm⁻². The cells were allowed to adhere overnight before imaging.

Primary HBMECs (Cell Systems), kept to a passage number lower than 10, were cultured in EGM-2 MV medium (Lonza) on flasks coated with poly-L-lysine (Sigma-Aldrich). HBMECs were then seeded onto dishes with 30 µl of 7.5 mg ml⁻¹ collagen I solution (isolated from rat tails) spread evenly. HBMECs were allowed to attach and spread on the collagen overnight before being imaged the next day.

To test for cell viability under low-power pulsed SBS and high-power CW SBS, we added PI (≥94%, P4170-10MG, Sigma-Aldrich) into the cell dishes with a volume ratio to the cell medium of 30 µl in 2 ml.

**Zebrafish.** A wild-type zebrafish larva (3 dpf) was used in Fig. 3a–g. 1-Phenyl 2-thiourea was added at 0.003% concentration at 12 h post fertilization to avoid pigmentation. The fish was mounted in the center of a Petri dish (Mattek, 35 mm dish, No. 0 coverslip) with 1% (w/v) low-melting agarose and 0.016% (w/v) tricaine (for immobilization purpose). A coverslip was used to cover the agarose before solidifying to allow the access of the sample by two objectives.

***C. elegans.*** *C. elegans adult preparation.* N2 Bristol animals, 6 h post-L4, were washed twice in M9 buffer (22 mM KH₂PO₄, 49 mM Na₂HPO₄, 86 mM NaCl and 10 mM NH₄Cl) with 0.1% (v/v) Tween-20 followed by two washes in M9 buffer with 0.1% (w/v) tetramisole (mounting solution). The animals were mounted in 5 µl of mounting solution between two agarose pads of 120 µm thickness each. To make agarose pads, 18 µl freshly prepared 5% (w/v) low-melting agarose in M9 buffer with 0.01% (w/v) tetramisole were pipetted onto a glass slide and pressed against another slide. To control for the agarose pad thickness, a 120-µm spacer was placed on the slide containing the agarose. The final sample thickness was ~240 µm.

*C. elegans embryo preparation. C. elegans* adults of the N2 Bristol strain were isolated on agar plates and allowed to lay eggs, corresponding to embryos of 50–100 cell stage and 100–130 min post-fertilization. Two hours post-egg-laying, embryos were transferred using platinum wire onto glass slides with M9 solution. Embryos were washed five times using transfer pipetting into M9 solutions, to remove bacteria before final mounting. These embryos of 'late ball stage' (approximately 260–300 min post-fertilization) were mounted on a 2% agarose pad in H₂O solution. Twenty-millimeter polystyrene beads were used as slide spacers in the sample preparation. This mounting causes stereotypical turns of the embryos in the eggshell during morphogenesis due to slight compression. Agarose pad was prepared using a 120-µm spacer on top of the slide to control the thickness. The final sample thickness was ~180 µm. Glass slides were covered with coverslips and sealed using liquid Vaseline heated to 40 °C, to prevent H₂O evaporation. Brightfield and Brillouin images were acquired in 13-min intervals.

**Mouse organoids.** Breeding and maintenance of the mouse colony was done in the Laboratory Animal Resources facility of EMBL Heidelberg. Primary mammary epithelial cells were obtained from 8-week-old virgin females of Friend Virus B (FVB) background. Three-dimensional cell cultures were established according to the published protocol[30,47].

In short, the tissue from two mammary glands without mechanical dissociation was placed in 5-ml digestion medium (DMEM/F12 with L-glutamine, 15 mM HEPES (Biowhittaker, no. 12-719Q), supplemented with 1 M HEPES (Biowhittaker, no. 17-737E) to 25 mM final concentration), 150 U ml⁻¹ collagenase type 3 (Worthington, no. CLS3) and 20 µg ml⁻¹ Liberase Blendzyme 2 (Roche, no. 11988425001), and digested for 15–16 h at 37 °C in loosely capped 50-ml polypropylene conical tubes. The resultant organoid suspension was washed with 45 ml of PBS containing Ca⁺⁺ and Mg⁺⁺, pelleted at 1,000 rpm for 5 min at room temperature, and resuspended in 5 ml of 0.25% trypsin–ethylenediaminetetraacetic acid. After incubation for 40 min at 37 °C in loosely capped 50-ml polypropylene conical tubes, cells were washed with 45 ml of DMEM/F12 with L-glutamine, 15 mM HEPES, supplemented with 1 M HEPES to 25 mM final concentration and with 10% Tet System Approved FBS (Clontech, no. 631101). Suspensions were treated with 10 mg ml⁻¹ DNaseI (Sigma, no. D4527). Dissociated cells were pelleted at 1,000 rpm for 5 min at room temperature and resuspended in PBS containing Ca⁺⁺ and Mg⁺⁺, counted, and plated onto collagen-coated 10-cm plates (BioCoat, #356450) for overnight adhesion and expansion. Then cultured cells were washed with PBS without Ca⁺⁺ and Mg⁺⁺, and the remaining cells were treated with 1 ml 0.25% trypsin–ethylenediaminetetraacetic acid. After cell detachment, trypsin was inactivated with 9 ml DMEM/F12 with L-glutamine, 25 mM HEPES, supplemented with 10% Tet System Approved FBS. Cells were pelleted at 1,000 rpm for 5 min at room temperature and resuspended in PBS containing Ca⁺⁺ and Mg⁺⁺ and counted. Cells were mixed rapidly on ice with Matrigel Matrix (Corning, 354230) Droplets (100 µl) containing 10,000 cells from mammary gland preparations and were dispensed in a Petri dish (Mattek, 35 mm dish, No. 0 coverslip) with a well in the center. After solidification on a level surface for 30 min at 37 °C, the gels were placed at 37 °C in a CO₂ incubator with 1.5 ml of supplemented serum-free medium (500 ml MEBM mammary epithelial cell growth basal medium (Lonza, CC-3151) supplemented with 2 ml of bovine pituitary extract, 0.5 ml of hEGF, 0.5 ml of hydrocortisone, 0.5 ml of GA-1000 and 0.5 ml insulin from MEGM mammary epithelial cell growth medium BulletKit (Lonza CC-3150)). Medium was replaced every 2 days. A coverslip was used to cover the cell medium from evaporation and to allow access of the sample by two objectives.

**Mouse strains and embryo culture.** All mice used in this study were housed according to the guidelines of EMBL Laboratory Animal Resources. Mice were maintained in individually ventilated plastic cages (Tecniplast) in an air-conditioned (temperature 22 ± 2 °C, humidity 50 ± 10%) and light-controlled room (illuminated from 7:00 to 19:00). Mice were fed 1318P autoclavable diet (Altromin) ad libitum. Mouse embryos were collected from superovulated (injection of 7.5 international units (IU) pregnant mare serum followed by injection of 7.5 IU human chorionic gonadotropin 42–50 h later) 8- to 24-week-old female mice according to the guidelines of EMBL Laboratory Animal Resources. Embryos were cultured in 30-µl drops of G1 PLUS (Vitrolife) covered by mineral oil (Ovoil, Vitrolife). Embryos were isolated from C57BL/6J × C3H/He F1 females.

### Reporting summary

Further information on research design is available in the Nature Portfolio Reporting Summary linked to this article.

## Data availability

The raw datasets generated and/or analyzed during the current study are available at https://doi.org/10.5281/zenodo.8211867. Source data are provided with this paper.

## Code availability

The GPU-accelerated SBS spectral analysis code is open source under a GPLv3 license and can be accessed at https://github.com/prevedel-lab/sbs-gpu-acceleration.

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

## Acknowledgements

We thank the mechanical and electronic workshops at EMBL Heidelberg for help; A. Bilenca for helpful advice during early stages of the project; S. Möllmert and D. Wehner (MPL Erlangen) for discussions and expert advice; M. Schindler and N. Petridou (EMBL) for providing zebrafish larvae. M.G. was supported by a research fellowship from the EMBL Interdisciplinary Postdoc (EIPOD) Programme under Marie Skłodowska Curie Cofund Actions MSCA-COFUND-FP grant (847543). A.D.-M. was supported by the Deutsche Forschungsgemeinschaft (DFG) research grants DI 2205/2-1 and DI 2205/3-1. R.P. acknowledges support of an ERC Consolidator Grant (no. 864027, Brillouin4Life), the German Center for Lung Research (DZL), research funding 'Life Science' of the Molit Institute, and a COST Innovator grant (IG16124). R.P. and A. D.-M. acknowledge funding from the COST Action CA16124 ('BioBrillouin'). This work was supported by the European Molecular Biology Laboratory.

## Author contributions

F. Y. and R.P. conceived the project. F.Y. and R.P. designed the imaging system, and F.Y. realized it with the help from C.B. S.H. wrote the custom spectral analysis code. F.Y. performed experiments and analyzed data, with the help from G.R. and M.J., as well as from A.N., A.G., K.W. and M.G., under the guidance of S.K., A.D.-M., J.E. and M.B., respectively. R.P. led the project and wrote the paper together with F.Y. and input from all authors.

## Funding

## Competing interests

M.J. is an employee of MOLIT Institute for Personalized Medicine gGmbH. All other authors declare no competing interests.

## Additional information

**Extended data** is available for this paper at https://doi.org/10.1038/s41592-023-02054-z.

**Correspondence and requests for materials** should be addressed to Fan Yang or Robert Prevedel.

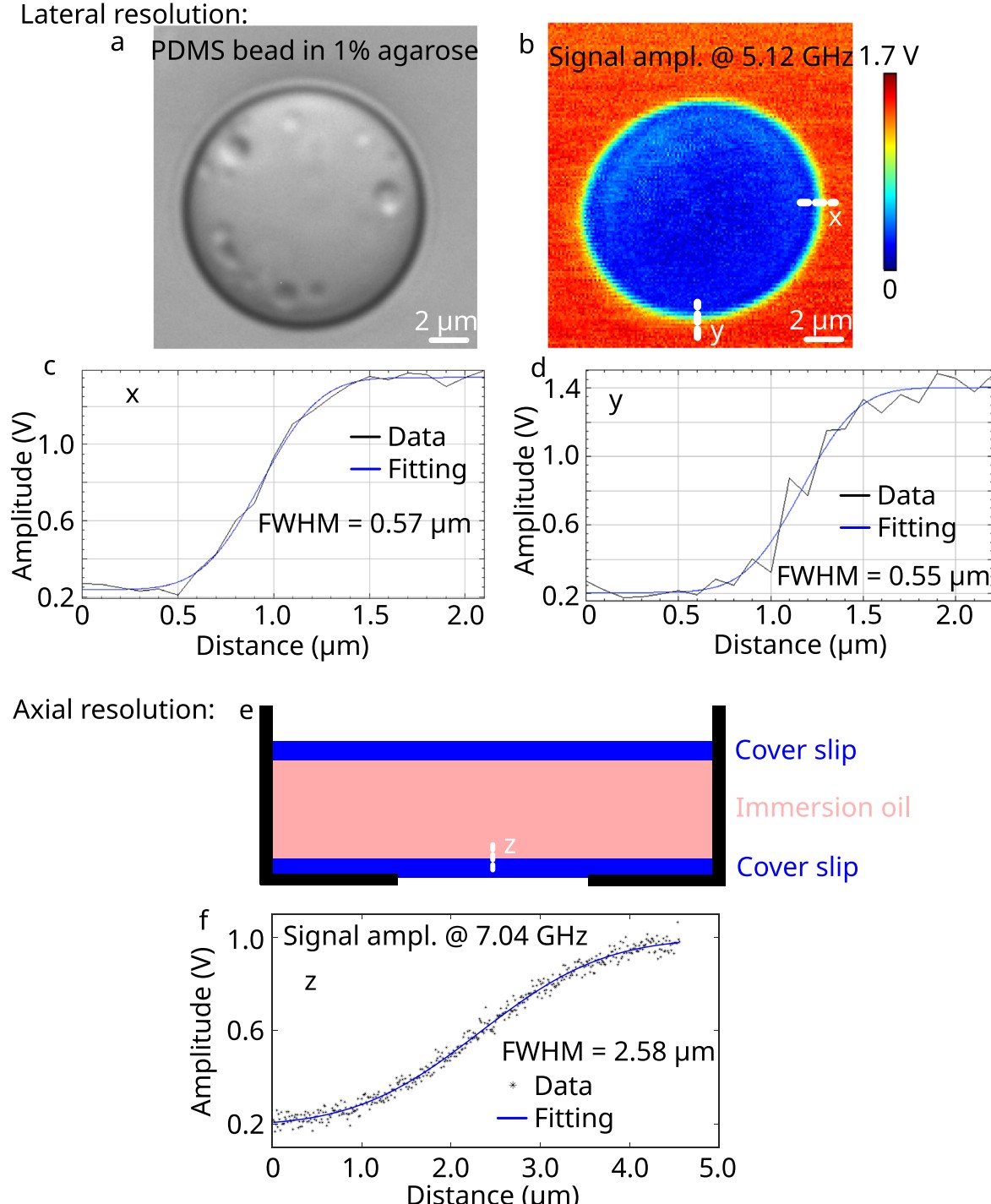

**Extended Data Fig. 1 | Characterization of the lateral (x and y) and axial (z) resolution of the pulsed-SBS microscope.** The lateral resolution (x and y) and axial resolution (z) were measured with a PDMS bead in 1% agarose and immersion oil sandwiched by two cover slips, respectively. (**a**) Brightfield image of a PDMS bead in 1% agarose. (**b**) Fitted signal amplitude at 5.12 GHz which is the Brillouin shift of 1% agarose. (**c**) and (**d**) are the amplitudes of Lorentzian fits to the spectrum, which are themselves fitted with an erf function at 5.12 GHz along the edge in the x and y direction, respectively. (**e**) Schematic diagram of immersion oil sandwiched between two coverslips, with the white line indicating the measurement scan. (**f**) The amplitude of Lorentzian fits to the spectrum at 7.04 GHz, which is the Brillouin shift of the immersion oil, and an erf-function fit to the data.

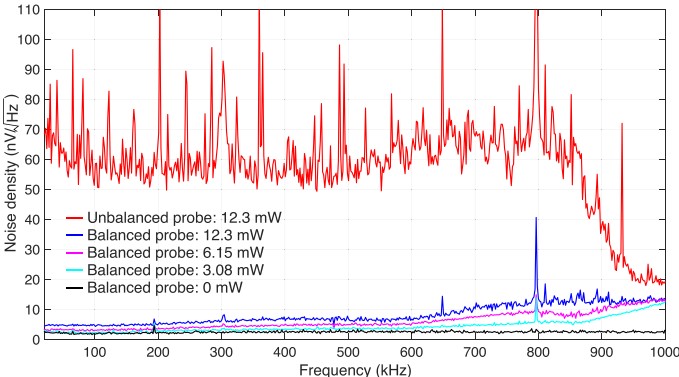

**Extended Data Fig. 2 | Balanced vs. unbalanced detection performance of pulsed-SBS.** The plot shows the noise density against the frequency for different incident powers on the photodiode. Note that the pump amplitude envelope modulation frequency is selected at 320 kHz in our experiments.

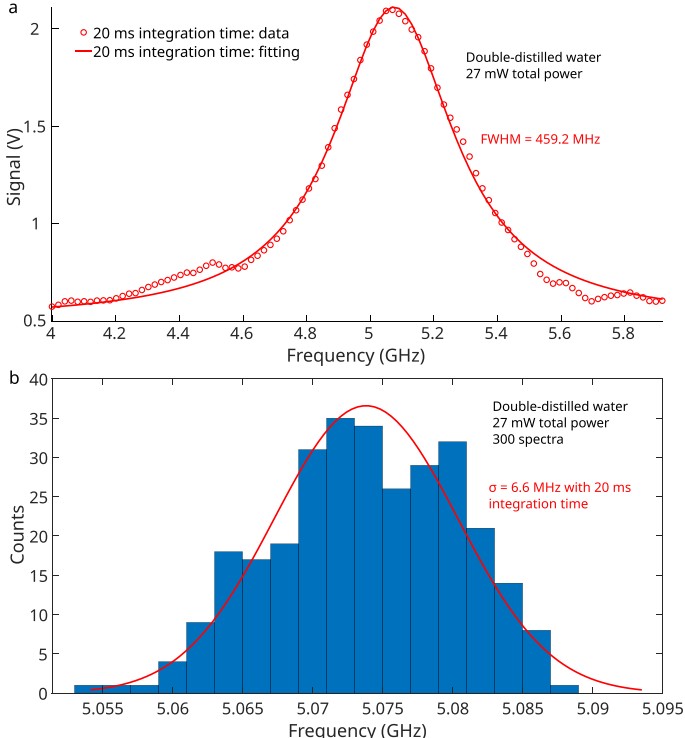

**Extended Data Fig. 3 | SBG spectrum and measurement precision of water. (a)** A representative SBG spectrum of water with 20 ms integration time and 27 mW total power. **(b)** Brillouin shift precision, obtained from 300 sequential measurements of water with 20 ms integration time and 27 mW total power. These constitute the same parameters as used for most biological samples imaged in this work.

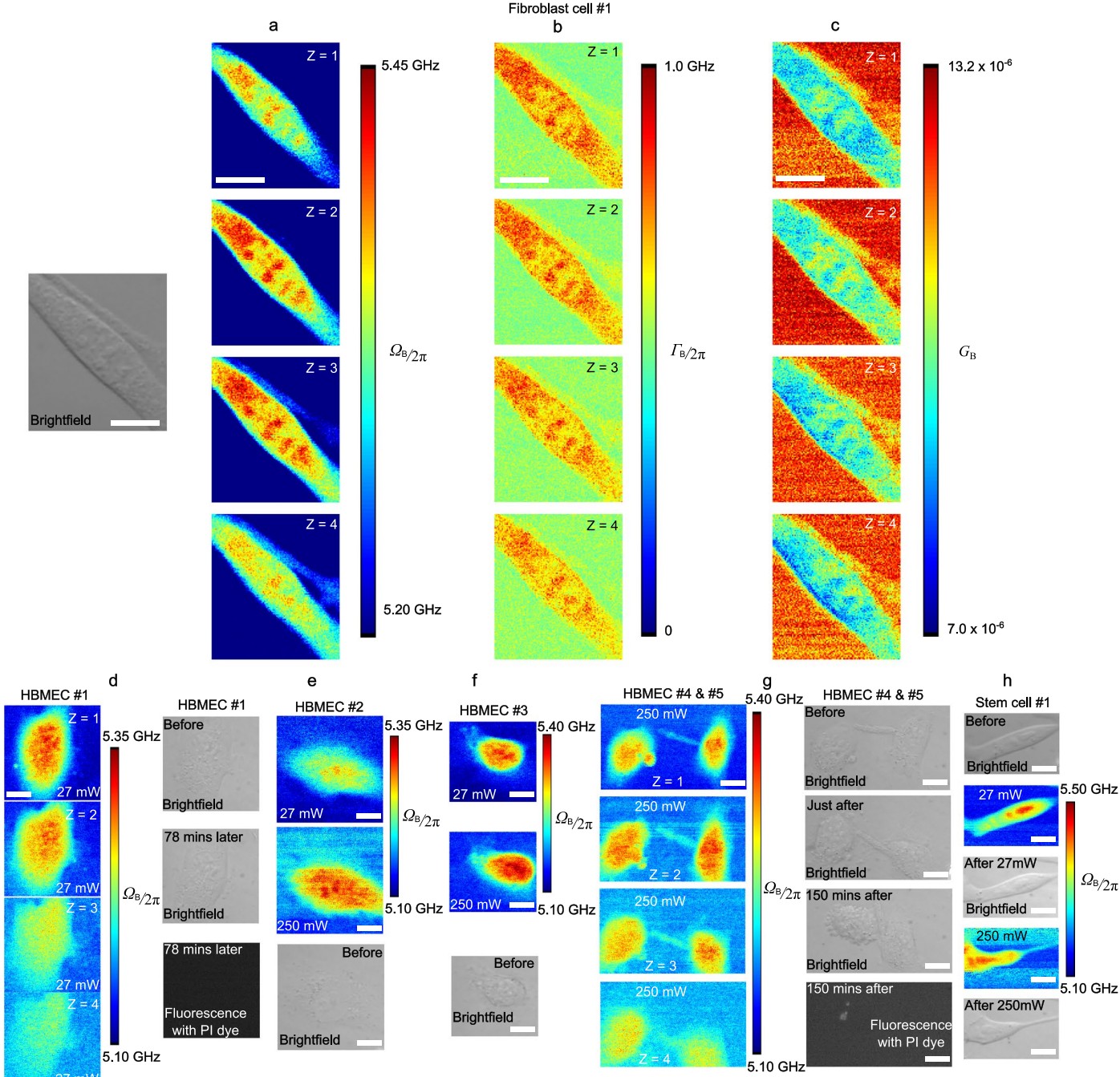

**Extended Data Fig. 4 | Representative 3D SBS images of a mouse fibroblast cell, human brain microvascular endothelial cells (HBMEC) and a mouse embryonic stem cell.** Three-dimensional sections of Brillouin shift (**a**), Brillouin linewidth (**b**) and Brillouin gain (**c**) images of a fibroblast cell under 20 mW pump and 7 mW probe with a z step of 1 µm. (**d**) Three-dimensional sections of Brillouin shift images of a HBMEC under 20 mW pump and 7 mW probe with a z step of 1 µm. PI, propidium iodine (viability marker that indicates damaged membranes). Brillouin shift images of HBMEC #2 (**e**) and HBMEC #3 (**f**) under 27 mW total power and 250 mW total power and the co-registered brightfield image. The overall Brillouin shift of the two cells were elevated due to high power as quantified in Fig. 2j. (**g**) Three-dimensional sections of Brillouin shift images of HBMEC #4 and #5 under 250 mW total power with a z step of 1 um. (**h**) A mouse stem cell was sequentially imaged with 27 mW pulsed and 250 mW CW schemes. Scale bar: 10 µm.

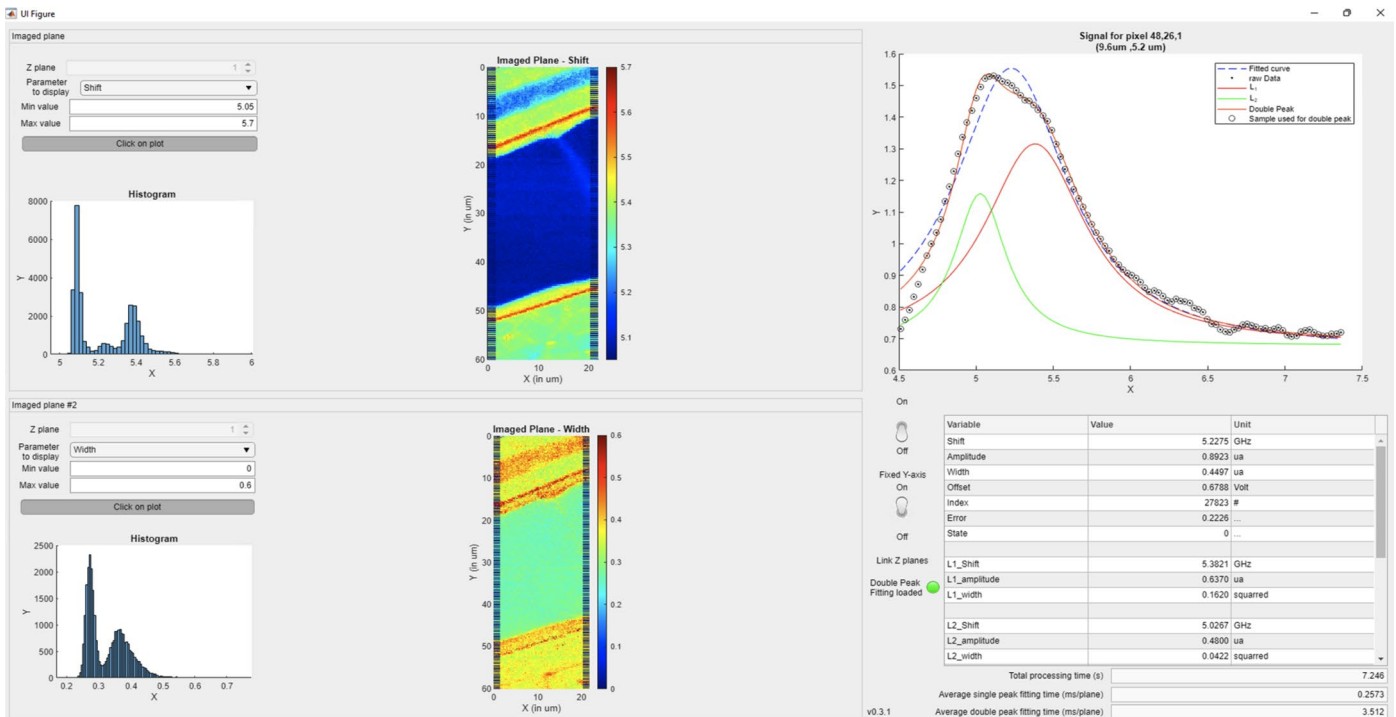

**Extended Data Fig. 5 | Graphic user interface (GUI) of SBS spectral analysis.**
The home-made GUI includes the parameter selection panel, 2 image panels after GPU fitting, histogram of the fitted parameter in the whole image, the raw data spectrum, the single peak fitting spectrum and the double peak fitting spectra as well as the fitted parameters at a specific pixel selected by the user. As an example, the pixel in the spinal chord central canal region at x = 9.6 μm, y = 5.2 μm position was selected and its SBG spectrum and the double-peak fitting results are shown in the GUI.

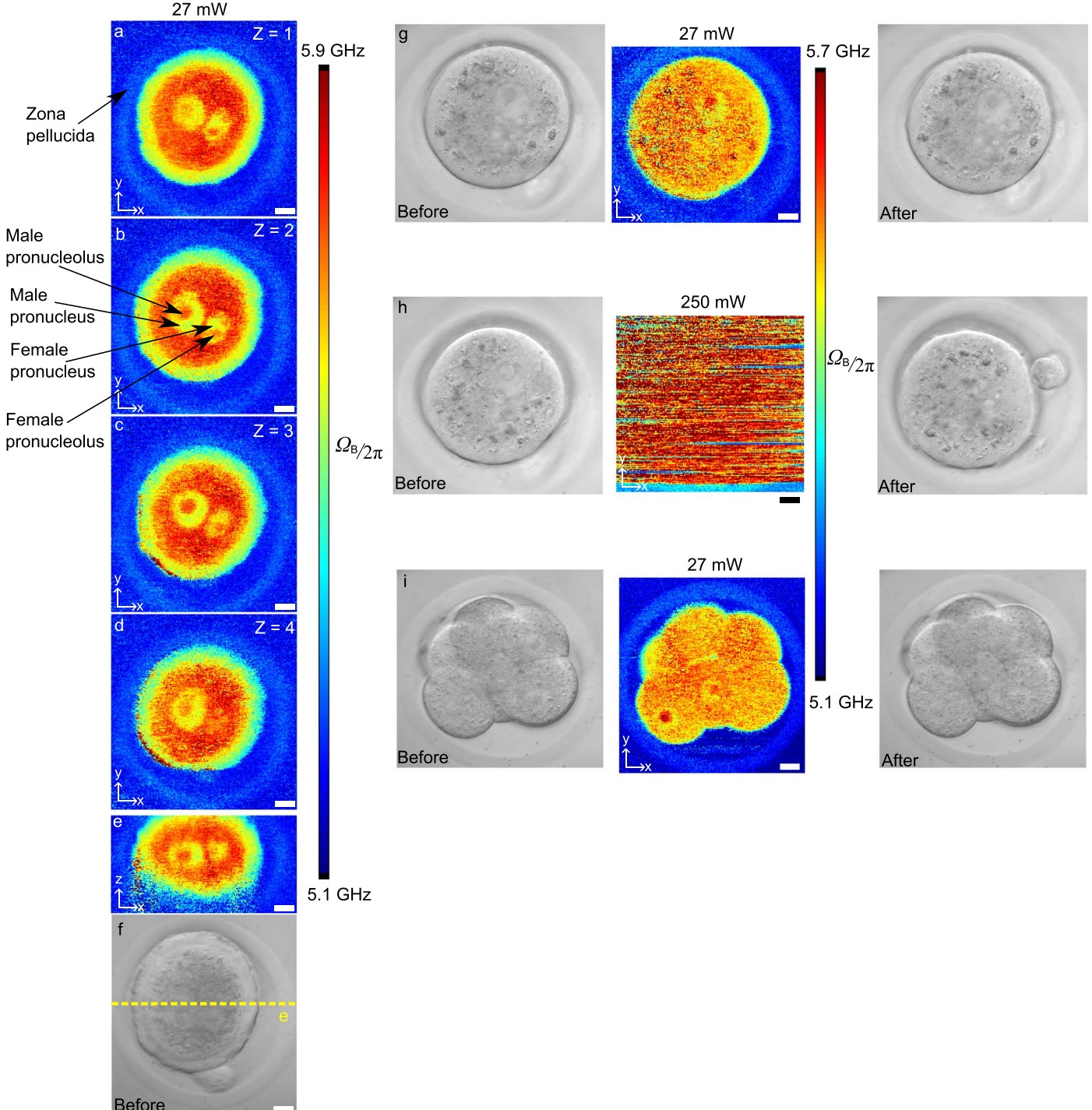

**Extended Data Fig. 6 | SBS imaging of mouse embryos.** Three-dimensional x-y cross sections (**a**)-(**d**) with a z step of 3 μm and (**e**) x-z cross sectional Brillouin images of a mouse zygote. (**f**) Brightfield image of the zygote. (**g**) Brillouin and brightfield images of a zygote under a 27 mW pulsed scheme. (**h**) Brillouin and brightfield images of a zygote under a 250 mW a CW scheme. Note that the high CW laser power trapped the zygote in the focus during the 2D piezo scanning. Therefore, it was not possible to obtain a Brillouin map of the embryo with the CW scheme. (**i**) Brillouin and brightfield images of an 8-cell-stage mouse embryo. Scale bar: 10 μm.

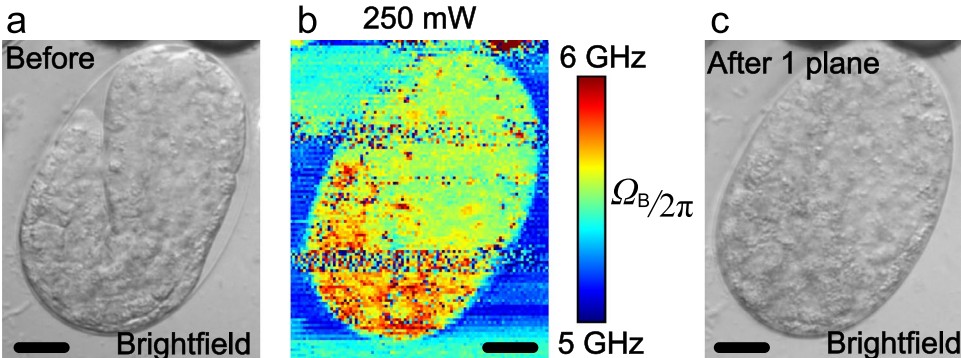

**Extended Data Fig. 7 | Photodamage after CW-SBS imaging of *C. elegans* embryo.** Brightfield images of a *C. elegans* embryo before (**a**) and after (**c**) a single plane Brillouin image acquisition. (**b**) Brillouin image with 250 mW power in the CW-SBS scheme. Scale bar: 10 μm.

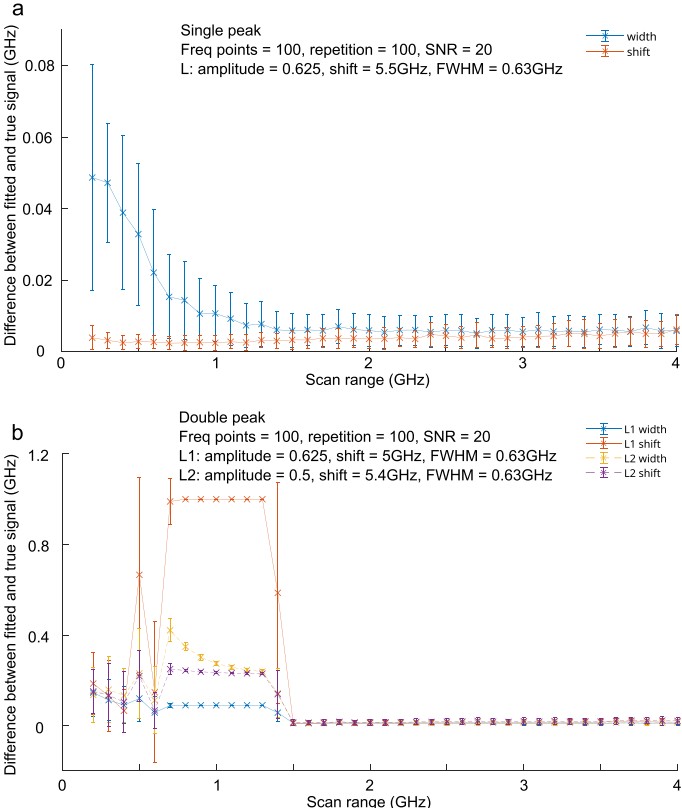

**Extended Data Fig. 8 | Simulation of pulsed-SBS spectral accuracy under different scan ranges. (a)** The Brillouin shift and linewidth accuracy of single-peak fitting as a function of scan range. The signal to be fitted is the simulated sum of a Lorentzian (with an amplitude of 0.625, peak shift 5.5 GHz, FWHM 0.63 GHz) and a random noise with a standard deviation amplitude of 0.0312 (that is SNR of 20). **(b)** The Brillouin shift and linewidth accuracy of a double-peak fitting as a function of scan range. The signal to be fitted is the sum of two Lorentzian waveforms (L1 with an amplitude of 0.625, peak shift 5 GHz, FWHM 0.63 GHz, L2 with an amplitude of 0.5, peak shift 5.4 GHz, FWHM 0.63 GHz) and a random noise with a standard deviation of 0.0312 (that is SNR of 20). There are 100 frequency points in the scans, to mimic the actual experimentally acquired spectra. The accuracy and error bar are the mean and standard deviation of 100 independent simulations, respectively. Note that above 1.5 GHz, the double peak is fitted with L1 and L2, but between 0.8 and 1.5 GHz, only L2 gets a meaningful fit and L1 is minimized because it doesn't recognize the second peak.

# Reporting Summary

Nature Research wishes to improve the reproducibility of the work that we publish. This form provides structure for consistency and transparency in reporting. For further information on Nature Research policies, see Authors & Referees and the Editorial Policy Checklist.

## Statistical parameters

When statistical analyses are reported, confirm that the following items are present in the relevant location (e.g. figure legend, table legend, main text, or Methods section).

| n/a | Confirmed | |
|---|---|---|
| ☐ | ☒ | The exact sample size (*n*) for each experimental group/condition, given as a discrete number and unit of measurement |
| ☐ | ☒ | An indication of whether measurements were taken from distinct samples or whether the same sample was measured repeatedly |
| ☐ | ☒ | The statistical test(s) used AND whether they are one- or two-sided<br>*Only common tests should be described solely by name; describe more complex techniques in the Methods section.* |
| ☒ | ☐ | A description of all covariates tested |
| ☒ | ☐ | A description of any assumptions or corrections, such as tests of normality and adjustment for multiple comparisons |
| ☐ | ☒ | A full description of the statistics including central tendency (e.g. means) or other basic estimates (e.g. regression coefficient) AND variation (e.g. standard deviation) or associated estimates of uncertainty (e.g. confidence intervals) |
| ☐ | ☒ | For null hypothesis testing, the test statistic (e.g. *F*, *t*, *r*) with confidence intervals, effect sizes, degrees of freedom and *P* value noted<br>*Give P values as exact values whenever suitable.* |
| ☒ | ☐ | For Bayesian analysis, information on the choice of priors and Markov chain Monte Carlo settings |
| ☒ | ☐ | For hierarchical and complex designs, identification of the appropriate level for tests and full reporting of outcomes |
| ☒ | ☐ | Estimates of effect sizes (e.g. Cohen's *d*, Pearson's *r*), indicating how they were calculated |
| ☐ | ☒ | Clearly defined error bars<br>*State explicitly what error bars represent (e.g. SD, SE, CI)* |

*Our web collection on statistics for biologists may be useful.*

## Software and code

Policy information about availability of computer code

| Data collection | Custom written LabView 2018 control software. |
|---|---|
| Data analysis | Custom written Matlab R2021b scripts, CUDA 11.5 code (see https://github.com/prevedel-lab/sbs-gpu-acceleration), GPUfit v1.2.0; Fiji 1.52i |

For manuscripts utilizing custom algorithms or software that are central to the research but not yet described in published literature, software must be made available to editors/reviewers upon request. We strongly encourage code deposition in a community repository (e.g. GitHub). See the Nature Research guidelines for submitting code & software for further information.

## Data

Policy information about availability of data

All manuscripts must include a data availability statement. This statement should provide the following information, where applicable:
- Accession codes, unique identifiers, or web links for publicly available datasets
- A list of figures that have associated raw data
- A description of any restrictions on data availability

The raw datasets generated and/or analysed during the current study are available at https://doi.org/10.5281/zenodo.8211867. Source data are provided with this paper.

# Field-specific reporting

Please select the best fit for your research. If you are not sure, read the appropriate sections before making your selection.

☒ Life sciences          ☐ Behavioural & social sciences          ☐ Ecological, evolutionary & environmental sciences

For a reference copy of the document with all sections, see nature.com/authors/policies/ReportingSummary-flat.pdf

# Life sciences study design

All studies must disclose on these points even when the disclosure is negative.

| | |
|---|---|
| Sample size | This work focused on the development of a new imaging technique. The performance of the microscope was validated on imaging double-distilled water, mouse fibroblast cells, primary human brain microvascular endothelial cells, mouse stem cells, young adult C. elegans worms, zebrafish larvae, C. elegans embryos, mouse embryos and mouse mammary gland organoids. In general, the imaging was repeated between 2 to 30 times on each cell, embryo or organism, all of which produced comparable data quality. These numbers are indicated in the main text and below. This sample size was sufficient in our opinion since we were demonstrating a microscope's performance and were not investigating a biological question. For microscope characterization, a large spectral sample size (n=300) was chosen to obtain Normal distribution. |
| Data exclusions | No data was intentionally excluded from the study. Representative data sets were chosen for the Figures. |
| Replication | We repeated the in-vivo imaging experiments multiple times on a total of n=30 individual fibroblast cells, n=5 primary human brain microvascular endothelial cells, n=2 mouse stem cells, n=10 C.elegans worms, n=4 zebrafish larvae, n=5 mouse mammary gland organoids, n=9 C.elegans embryos, and n=10 mouse embryos. The results were reproducible, i.e. they yielded image datasets of comparable quality. |
| Randomization | We did not preselect or pre-screen samples, but chose them randomly in order to demonstrate the capabilities of our new microscopy technique. |
| Blinding | In principle we were blinded to any group allocation and the outcome of our study, i.e. the demonstration of a new microscopy technique, is independent of the biological sample studied. |

# Reporting for specific materials, systems and methods

## Materials & experimental systems

| n/a | Involved in the study |
|---|---|
| ☒ | ☐ Unique biological materials |
| ☒ | ☐ Antibodies |
| ☐ | ☒ Eukaryotic cell lines |
| ☒ | ☐ Palaeontology |
| ☐ | ☒ Animals and other organisms |
| ☒ | ☐ Human research participants |

## Methods

| n/a | Involved in the study |
|---|---|
| ☒ | ☐ ChIP-seq |
| ☒ | ☐ Flow cytometry |
| ☒ | ☐ MRI-based neuroimaging |

# Eukaryotic cell lines

Policy information about cell lines

| | |
|---|---|
| Cell line source(s) | Mouse fibroblast cells: NIH/3T3-CRT-1658 from ATCC; Mouse embryonic stem cells: sox1-GFP-mESCs from Austin Smith's lab, University of Exeter; Primary human brain microvascular endothelial cells: HBMECs from Cell systems. |
| Authentication | These cell lines have been authenticated by the vendors. In particular, we purchased the NIH/3T3 cells from ATCC where they were authenticated by cytochrome 1 oxidase barcoding for species detection and morphology. The cells have also been tested for mycoplasma and ectromelia virus and were confirmed to be negative for both. The HBMECs from Cell systems were validated by immunofluorescence for endothelial markers. |
| Mycoplasma contamination | The cell lines used in this study are not contaminated by mycoplasma. |
| Commonly misidentified lines (See ICLAC register) | There is no misidentified cell line used in this study. |

# Animals and other organisms

Policy information about studies involving animals; ARRIVE guidelines recommended for reporting animal research

Laboratory animals

This work followed the European Communities Council Directive (2010/63/EU) to minimize animal pain and discomfort. All animal care and procedures performed in this study conformed to the EMBL Guidelines for the Use of Animals in Experiments and were reviewed and approved by the Institutional Animal Care and Use Committee (IACUC), under protocol number 2020-01-06RP. (C57BL/6JxC3H/He) F1 mice from eight-weeks of age onwards were used. Embryos were recovered from superovulated female mice mated with male mice. Mice were maintained in individually ventilated plastic cages (Tecniplast) in an air-conditioned (temperature 22 °C ± 2 °C, humidity 50% ± 10%) and light-controlled room (illuminated from 07:00 to 19:00 h). Mice were fed 1318 P autoclavable diet (Altromin, Germany) ad libitum.

Wild-type zebrafish larvae (3 days after fertilization) were used in the experiments.
N2 Bristol strain C. elegans worms (6 hours post L4 stage) were used in the C. elegans worm experiments. The C. elegans embryos (2 hours post-egg-laying) were transferred using platinum wire onto glass slides with M9 solution.

Wild animals

The study did not involve wild animals.

Field-collected samples

The study did not involve field-collected samples.

