## [Peer Review File · Nature Methods]

Peer Review Information

Manuscript Title: Pulsed stimulated Brillouin microscopy enables high-sensitivity mechanical imaging of live and fragile biological specimens

Corresponding author name(s): Robert Prevedel

Editorial Notes: n/a

Reviewer Comments & Decisions:

Decision Letter, initial version:

Dear Robert,

I am very sorry about the delays regarding your manuscript. We were waiting to hear back from one more reviewer, but unfortunately, they did not deliver as promised.

Your Article, "Pulsed stimulated Brillouin microscopy enables high-sensitivity mechanical imaging of live and fragile biological specimens", has now been seen by two reviewers. As you will see from their comments below, although the reviewers find your work of considerable potential interest, they have raised a number of concerns. We are interested in the possibility of publishing your paper in Nature Methods, but would like to consider your response to these concerns before we reach a final decision on publication.

We therefore invite you to revise your manuscript to address these concerns. In particular, we ask that the limitations of the method are clearly stated.

* include a point-by-point response to the reviewers and to any editorial suggestions

* please underline/highlight any additions to the text or areas with other significant changes to facilitate review of the revised manuscript

* address the points listed described below to conform to our open science requirements

* ensure it complies with our general format requirements as set out in our guide to authors at www.nature.com/naturemethods

* resubmit all the necessary files electronically by using the link below to access your home page

[Redacted]

This URL links to your confidential home page and associated information about manuscripts you may have submitted, or that you are reviewing for us. If you wish to forward this email to co-authors, please delete the link to your homepage.

We hope to receive your revised paper within four weeks. If you cannot send it within this time, please let us know. In this event, we will still be happy to reconsider your paper at a later date so long as nothing similar has been accepted for publication at Nature Methods or published elsewhere.

OPEN SCIENCE REQUIREMENTS

REPORTING SUMMARY AND EDITORIAL POLICY CHECKLISTS

Please note that these forms are dynamic ‘smart pdfs’ and must therefore be downloaded and completed in Adobe Reader. We will then flatten them for ease of use by the reviewers. If you would like to reference the guidance text as you complete the template, please access these flattened versions at <http://www.nature.com/authors/policies/availability.html>.

DATA AVAILABILITY

We strongly encourage you to deposit all new data associated with the paper in a persistent repository where they can be freely and enduringly accessed. We recommend submitting the data to discipline-specific and community-recognized repositories; a list of repositories is provided here:

<http://www.nature.com/sdata/policies/repositories>

All novel DNA and RNA sequencing data, protein sequences, genetic polymorphisms, linked genotype and phenotype data, gene expression data, macromolecular structures, and proteomics data must be deposited in a publicly accessible database, and accession codes and associated hyperlinks must be provided in the “Data Availability” section.

Please include a “Data availability” subsection in the Online Methods. This section should inform readers about the availability of the data used to support the conclusions of your study, including accession codes to public repositories, references to source data that may be published alongside the paper, unique identifiers such as URLs to data repository entries, or data set DOIs, and any other statement about data availability. At a minimum, you should include the following statement: “The data that support the findings of this study are available from the corresponding author upon request”, describing which data is available upon request and mentioning any restrictions on availability. If DOIs are provided, please include these in the Reference list (authors, title, publisher (repository name), identifier, year). For more guidance on how to write this section please see: <http://www.nature.com/authors/policies/data/data-availability-statements-data-citations.pdf>

CODE AVAILABILITY

Please include a “Code Availability” subsection in the Online Methods which details how your custom code is made available. Only in rare cases (where code is not central to the main conclusions of the paper) is the statement “available upon request” allowed (and reasons should be specified).

MATERIALS AVAILABILITY

ORCID

Best regards,
Nina

Nina Vogt, PhD
Senior Editor
Nature Methods

Reviewers' Comments:

Reviewer #1:
None

Reviewer #2:

Remarks to the Author:

A. Key result: achieved a 20x lower illumination power than CW-based SBS implementation using a novel quasi-pulsed approach

B. Original and novel work that constitutes an advancement on current CW SBS implementations in terms of laser power

C. High quality data and presentation, although the use of Brillouin shift as a proxy for stiffness or rigidity is not justified. All statements that refer to an increased shift as to an increased stiffness or

rigidity need to be addressed. Same goes for Brillouin linewidth and viscosity. A reference to the effects of density and refractive index gradients needs to be made, at the very least. Also, "mechanical specificity" is not the right wording throughout, since it is just spectral resolution (it doesn't refer to the Brillouin signal itself). The observation of multiple Brillouin peaks in a material layer that is thinner than the acoustic phonon wavelength requires attention. What are the hybrid acoustic modes introduced to interpret Fig.3? Red blood cells having similar Brillouin shift as the ECM needs to be justified. Fig.4h-j shows some stripes in the inner space, what is the origin of those? Discussion states "biological samples previously inaccessible to SBS", it can't be right.

D. All is good.

E. Conclusions are supported as long as point C is addressed in full.

F. Revision will need to address all points in C above.

G. All seems to be in order.

H. A minor point is that in the title and abstract "pulsed" is used, and then "quasi-pulsed". The two wordings need to match.

Reviewer #3:

Remarks to the Author:

The development of non-destructive, label-free and contact-free techniques for mechanobiology is very emerging field in scientific community. Mechanical properties of biological materials were proven to play a key role in many cellular/pathological processes and mechanomarkers were proposed as new diagnostic parameters to support current diagnostic methods. The manuscript by Yang et al., shows another modification of Brillouin microscopy that allows for live imaging of the mechanical properties of soft biological materials. The proposed implementation allows to decrease pump laser power to the extend where live cell/organoids measurements are possible with significant reduction of phototoxic effects that are otherwise observed in conventional laser illumination. The quality of presented data is very high with robust description and explanation. Text and figures are clear, even the supplementary information and video are refined and sufficiently explained. But prior I recommend the manuscript publication I would like to clarify some issues and suggest a couple of improvements.

E.g. Figure 2 – measurements of fibroblast cells on 12kPa PAA gel. It is not clear to me what is the source of the contrast between the cell and hydrogel in Brillouin shift/linewidth images. The stiffness of polyacrylamide and agarose is usually in the kPa – tens of kPa range, and according to early works of Janmey, Discher, Chaudhuri etc. (cited in the manuscript) stiffness of the cells that sit on top of such

hydrogels is comparable with substrate stiffness (e.g. Solon et al., “Fibroblast adaptation and stiffness matching to soft elastic substrates” Biophysical Journal 2007). How to understand e.g. the image on Fig.2f where cell cross section in z direction is presented. Why Brillouin shift/linewidth (elasticity/viscosity) of the hydrogels is so low compare to cell? What is the source of the contrast between the cell and the gel? I suspect the water content is comparable. Please, elaborate on that, also in the context of other samples that were embedded in gels.

Except of phototoxicity and high powers needed to register sufficient Brillouin spectra, Brillouin microscopy suffers many limitations. e.g. in this setup the sample thickness and optical transparency needs to be very tightly controlled. I encourage the authors to face these limitations in the introduction section and through the manuscript. Otherwise one could have a feeling that SBS can be directly implemented in clinics for mechanotyping of biopsy specimens or other urgent application. I also suggest expanding the reference list, for instance the group of Prof. Jochen Guck publish a lot on BM measurements of biological specimens (like spinal cord development).

Overall, the manuscript is worth to be published if a deeper discussion on above mentioned could be carried on.

Author Rebuttal to Initial comments

Point-by-point reply for NMETH-A51474**Remark to all Reviewers:**

We would like to thank all Reviewers for their reports and valuable comments, which we believe, have helped to significantly enhance the quality of our manuscript. Below we give a point-by-point response to all issues that were raised and how we have addressed them in the revised version of our manuscript. To facilitate review of the revised manuscript, we have underlined any additions to the text or areas with other significant changes. We are confident that these improvements should fully address all Reviewers' comments and suggestions, and hope that it now meets their expectations.

Original Reviewer comments are in black. **Our replies are in blue.** *Changes to manuscript text are included in red where appropriate.*

Reviewer #2 (Remarks to the Author):

A. Key result: achieved a 20x lower illumination power than CW-based SBS implementation using a novel quasi-pulsed approach

B. Original and novel work that constitutes an advancement on current CW SBS implementations in terms of laser power

We thank the Reviewer for their positive assessment of our work!

C. High quality data and presentation, although the use of Brillouin shift as a proxy for stiffness or rigidity is not justified. All statements that refer to an increased shift as to an increased stiffness or rigidity need to be addressed. Same goes for Brillouin linewidth and viscosity. A reference to the effects of density and refractive index gradients needs to be made, at the very least.

We thank the Reviewer for this comment, and fully agree that additional information on the density and refractive index of the sample are required to deduce quantitative mechanical properties from Brillouin measurements. We have thus added clarifying sentences to the Introduction and throughout the paper, as well as have removed ambiguous statements towards the interpretation. In the revised introduction, we have also added two references (Ref. 10,11,12) which demonstrated the ratio ρ/n^2 does not vary significantly in the biological samples.

Changes to text:

The resulting Brillouin spectrum, i.e. the frequency shift, and linewidth of the inelastically scattered light then provides information on the longitudinal modulus (defined in the Online Methods) of the (bio-)material, assuming the relation between the refractive index and density is known. Although both the refractive index and the density may vary with conditions, their ratio ρ/n^2 does not vary significantly in biological materials¹⁰⁻¹². Therefore, the value of Brillouin frequency shift and linewidth are often reported as direct indicators of the mechanical properties.

10. Kim, K. & Guck, J. The Relative Densities of Cytoplasm and Nuclear Compartments Are Robust against Strong Perturbation. *Biophys. J.* 119, 1946–1957 (2020).

11. Scarcelli, G. et al. Noncontact three-dimensional mapping of intracellular hydromechanical properties by Brillouin microscopy. *Nat. Methods* 12, 1132–1134 (2015).

12. Schlüsler, R. et al. Mechanical Mapping of Spinal Cord Growth and Repair in Living Zebrafish Larvae by Brillouin Imaging. *Biophys. J.* 115, 911–923 (2018).

Also, "mechanical specificity" is not the right wording throughout, since it is just spectral resolution (it doesn't refer to the Brillouin signal itself).

We agree with the reviewer that the high 'specificity' in SBS is a direct consequence of its high spectral resolution. However, we note that previous seminal work in SBS (Remer *et al*, *Nat. Methods* 17, 913-916 (2020)) has coined this exact term, which we therefore also used in our manuscript. As noted in our reply (and the revised introduction), the higher spectral resolution implies mechanical specificity in the assumption that the ratio between refractive index and density does not significantly vary within cells and tissues.

Indeed, the higher spectral resolution of SBS (compared to VIPA based Brillouin spectrometers) enabled them to distinguish the pharynx in *C. Elegans* from the surrounding tissue, and hence to resolve the mechanical constituents that give rise to the overall spectrum. Similarly, we are able to distinguish up to 3 spectral peaks in the zebrafish larvae ECM (two acoustic ECM modes and the surrounding tissue).

To clarify this, and in line with our reply to the point above, we have added a sentence that elaborates on the connection between mechanical specificity and spectral resolution (see below). We hope the Reviewer agrees to this resolution.

Changes to text:

Here, the high spectral resolution of SBS reflects a high mechanical specificity²³ in the assumption that the ratio between refractive index and density does not significantly vary within cells and tissues, as previously shown in zebrafish¹².

23. Remer, I., Shaashoua, R., Shemesh, N., Ben-Zvi, A. & Bilenca, A. High-sensitivity and high-specificity biomechanical imaging by stimulated Brillouin scattering microscopy. Nat. Methods 17, 913–916 (2020).

12. Schlußler, R. et al. Mechanical Mapping of Spinal Cord Growth and Repair in Living Zebrafish Larvae by Brillouin Imaging. Biophys. J. 115, 911–923 (2018).

The observation of multiple Brillouin peaks in a material layer that is thinner than the acoustic phonon wavelength requires attention. What are the hybrid acoustic modes introduced to interpret Fig.3?

The reviewer raises a very interesting point here. Indeed, a single-peaked Brillouin spectrum splits to multiple peaks when the material layer is thinner than the optical wavelength. This is because when the material thickness is larger than the optical wavelength, the Brillouin signal results from scattering off the longitudinal acoustic mode. As the material thickness decreases and becomes thinner than the optical wavelength, light can not be fully confined in the bulk material and the pure longitudinal acoustic mode turns to hybrid acoustic modes which include both longitudinal and transverse motions. These hybrid modes generate multiple peaks. This phenomenon is observed and explained in micro and nano optical waveguide [Dainese *et al*, *Nature Physics* 2, 388-392 (2006); Beugnot *et al*, *Nature Communications* 5:5242 (2014)]. For example, Fig. R1 shows the Brillouin spectra for optical waveguides with core diameters of 1 μm and 9 μm at optical wavelength of 1.55 μm [Ref Dainese *et al*. *Nature Physics* 2, 388-392 (2006)]. When the core is much larger than the optical wavelength, the Brillouin spectrum shows only a single peak from the bulk material. When the core is smaller than the optical wavelength, it has multiple peaks.

In our case, the extracellular matrix (ECM) thickness of a 3dpf zebrafish larvae is ~ 400 nm [c.f. Bevilacqua *et al*, *Biomedical Optics Express* 10 1420-1431 (2019)], which is thinner than the optical wavelength used in our pulsed-SBS approach (780 nm). As stated in the manuscript and

shown in Fig. 3f, the measured Brillouin spectrum in the ECM region has three Brillouin peaks (L1 5.34 GHz, L2 5.63 GHz and L3 6.63 GHz). The L1 Brillouin shift is very close to the single Brillouin peak of the ECM-surrounding tissue (5.36 GHz) located just below the cross marker in Fig. 3c in the main manuscript. Therefore, we made the hypothesis that the right two peaks (i.e. L2 and L3 in Fig. 3f) are due to the hybrid acoustic modes in ECM. The hybrid acoustic modes have both longitudinal and transverse motions.

As suggested by the reviewer, we have clarified and elaborated on the origin of the hybrid acoustic modes in the revised manuscript.

Changes to text:

We hypothesize the L2 and L3 components to represent hybrid acoustic modes which involve both longitudinal and transverse motions, due to the ECM being thinner than the optical wavelength, since such hybrid acoustic mode-induced splitting is often observed in sub-wavelength optical waveguide²⁸.

28. Beugnot, J.-C. et al. Brillouin light scattering from surface acoustic waves in a subwavelength diameter optical fibre. Nat. Commun. 5, 5242 (2014).

[Redacted Third Party Material]

Fig. R1. Brillouin spectra for optical waveguides with core diameters of 1 μm (small core) and 9 μm (large core) at pump wavelength of 1.55 μm [Figure is taken from Fig. 1a of Reference: Dainese et al, Nature Physics 2, 388-392 (2006)].

Red blood cells having similar Brillouin shift as the ECM needs to be justified.

We note that both ECM modes (5.63 GHz and 6.63 GHz, see Fig. 3f) have indeed a higher Brillouin shift compared to the red blood cell (5.39 GHz, see Fig. 3g). The Brillouin shift values found for red blood cells are indeed in agreement with a previous report (Ref. Meng et al, *Journal of Biophotonics* 9, 201-207 (2016)). In this work, the authors measured bovine red blood cells with a Brillouin shift of 7.8 GHz at 532 nm, which corresponds to 5.32 GHz at 780 nm when taking the wavelength difference into account. We have clarified that L2 and L3 represent the ECM modes (see also reply above), and hope this addresses the Reviewer's comment.

Fig.4h-j shows some stripes in the inner space, what is the origin of those?

We thank the Reviewer for pointing this out. However, we would also like to note that we already included an explanation for this artefact in the caption of Fig. 4. in the original manuscript (line 521: "*particle trapping leads to stripe-like artefacts in the lumen of the organoids*"). Because of the optical tweezer effect, small (dust) particles can be trapped by the focused laser beams and are moved along inside the lumen of the organoids as the sample stage is scanning. This generates the stripe-like artefact. The same artefact can not be observed in the outside of the organoids as they are embedded in a matrigel which prohibits any particles from moving there. To further clarify this, we have moved the interpretation from the Fig. 4 caption to the revised main manuscript.

Changes to text:

Note that even with only 27 mW power, particle trapping due to optical tweezer effect leads to stripe-like artefacts in the lumen of the organoids shown in Fig. 4h-j.

Discussion states "biological samples previously inaccessible to SBS", it can't be right.

We thank the Reviewer for pointing this out. Indeed our intention was to state that many photosensitive samples could so far not be imaged *without photodamage* in SBS, and thus were 'inaccessible'. We realise our wording might be ambiguous, and have thus elaborated on our statement as summarised below:

Changes to text:

In our work we harnessed the improved efficiency of our approach along with diligent optimization of the signal detection to substantially reduce the required total illumination power and thus enable imaging of a wide range of photosensitive biological samples over extended time periods that could otherwise be damaged by the high laser powers required by previous SBS implementations.

A minor point is that in the title and abstract "pulsed" is used, and then "quasi-pulsed". The two wordings need to match.

We thank the Reviewer for pointing this out. To make it coherent, we decided to change "*quasi-pulsed*" to "*pulsed*" throughout the revised manuscript.

Reviewer #3 (Remarks to the Author):

The development of non-destructive, label-free and contact-free techniques for mechanobiology is very emerging field in scientific community. Mechanical properties of biological materials were proven to play a key role in many cellular/pathological processes and mechanomarkers were proposed as new diagnostic parameters to support current diagnostic methods. The manuscript by Yang et al., shows another modification of Brillouin microscopy that allows for live imaging of the mechanical properties of soft biological materials. The proposed implementation allows to decrease pump laser power to the extend where live cell/organoids measurements are possible with significant reduction of phototoxic effects that are otherwise observed in conventional laser illumination. The quality of presented data is very high with robust description and explanation. Text and figures are clear, even the supplementary information and video are refined and sufficiently explained. But prior I recommend the manuscript publication I would like to clarify some issues and suggest a couple of improvements.

We thank the Reviewer for their positive assessment of our work, as well as for their valuable comments, which we believe, have helped to significantly enhance the quality of our manuscript.

Figure 2 – measurements of fibroblast cells on 12kPa PAA gel. It is not clear to me what is the source of the contrast between the cell and hydrogel in Brillouin shift/linewidth images. The stiffness of polyacrylamide and agarose is usually in the kPa – tens of kPa range, and according to early works of Janmey, Discher, Chaudhuri etc. (cited in the manuscript) stiffness of the cells that sit on top of such hydrogels is comparable with substrate stiffness (e.g. Solon et al., "Fibroblast adaptation and stiffness matching to soft elastic substrates" *Biophysical Journal* 2007). How to understand e.g. the image on Fig.2f where cell cross section in z direction is presented. Why Brillouin shift/linewidth (elasticity/viscosity) of the hydrogels is so low compare to cell? What is the source of the contrast between the cell and the gel? I suspect the water content is comparable. Please, elaborate on that, also in the context of other samples that were embedded in gels.

The Reviewer raises an interesting point that we realize requires further elaborations. We fully agree with the Reviewer that it seems counterintuitive that a 12kPa PAA gel yields a lower Brillouin shift (and thus appear 'less stiff') than the cell. This is because of a combination of multiple reasons: First, in Brillouin microscopy, the measured shift is a proxy for the longitudinal storage modulus, which is not directly related to the Young's modulus or the shear modulus that is typically measured in mechanobiology (and used in the papers referenced by the Reviewer). Therefore, care must be taken when attempting to compare these fundamentally different, yet complementary, mechanical moduli (c.f. Box 4 in *Prevedel et al, Nat. Methods 16, 969-977 (2019)*). Second, typical rheological measurements (e.g. by AFM) measure the mechanical modulus at different frequencies (Hz-kHz for AFM, GHz for Brillouin), and the extrapolation from low frequency measurements to the high frequency regime is intrinsically material dependent. While empirical correlations have indeed been established for cells and tissues (see Fig. 1e in *Scarcelli et al, Nature Methods 12, 1132-1134 (2015)*; *Scarcelli et al, Biophys. J. 101, 1539-1545 (2011)*), care has to be taken for highly hydrated materials. In particular, in the GHz regime, the storage modulus becomes very sensitive to the water content (c.f. Fig. 2 in *Bailey et al, Science Advances 6 : eabc1937 (2020)*). In fact, PAA hydrogels possess an extremely high water fraction (~95-96% for 12kPa PAA, see *Wu et al, Nature Methods 15, 561-562 (2018)*), which is substantially different to cells and tissues (~70-75%, see Book: *G. M. Cooper, "The Cell: A Molecular Approach." 2nd edition, Sunderland (MA): Sinauer Associates (2000)*; *Scarcelli & Yun, Nature Methods 15, 562-563 (2018)*). This leads to the counterintuitive observation that hydrogels have a comparably low Brillouin shift (in fact, comparable to water), and we refer the Reviewer to the interesting discussion found in *Wu et al, Nature Methods 15, 561-562 (2018)* and *Scarcelli & Yun, Nature Methods 15, 562-563 (2018)* on that topic for further

information. We just highlight here that the Brillouin shift remains sensitive to solid constituents in less hydrated materials such as cells, and this is the source of contrast observed between the cell and gel in Fig. 2f. Likewise, the Matrigel used to embed the organoids (Fig. 4) possess ~99% water content, which is why the observed Brillouin shift of these gels is almost identical to the one for water (~5.08 GHz).

We further remark that other work in the field has previously imaged fibroblast cells (*Nikolic et al, Biomedical Optics Express 10, 1567-1580 (2019)*) or hydrogels (*Pahapale et al, Advanced Science 9, 202104649 (2022)*) with Brillouin microscopy (at 660nm), although no data exists in which both were imaged together. Still from those references we can extrapolate to our SBS measurements that are performed at 780nm. In Fig. 2f of our manuscript, the averaged Brillouin shifts are 5.50 GHz for the nucleolus, 5.41 GHz for the cytoplasm, 5.25 GHz for the 12kPa PAA gel and 5.12GHz for the cell medium respectively. In Ref. Nikolic 2019, the authors obtained 6.47GHz for the nucleolus and 6.15GHz for the cell medium at 660nm, which corresponds to 5.47GHz (nucleolus) and 5.20GHz (cell medium) at 780nm. Therefore, our measurements are in good agreement with this reference. In Ref. Pahapale 2022, the authors measured hydrogels with Brillouin shifts of 6.2 GHz for 2 kPa stiffness and 6.4 GHz for 35 kPa at a wavelength of 660nm. We linearly interpolate that a 12kPa hydrogel would have a Brillouin shift of ~6.26 GHz. Considering the wavelength, this corresponds to ~5.30 GHz which is also in good agreement with our measured 12kPa PAA gel Brillouin shift (5.25 GHz). Note in these two references, the authors used an objective with 60X, 0.7 NA which is very similar to the one used in our experiments. We thus find that the measured Brillouin shifts of both fibroblasts as well as hydrogels (interpolated to 12kPa) are in good agreement with our SBS results.

Furthermore, in order to confirm our findings obtained with our newly developed pulsed-SBS microscope, we have also imaged fibroblasts on a 12kPa PAA gel in a well-established, confocal Brillouin microscope at 532 nm (described in: *Bevilacqua et al, Biomedical Optics Express 10, 1420-1431 (2019)*). Fig. R2 below shows an exemplary cross-sectional image, which qualitatively further confirms our results in Fig. 2f.

To highlight the topic for the non-specialist reader without disrupting the main flow and message of the manuscript, we have included the following statements in the discussion section of the revised manuscript:

Changes to text:

We note that the high-frequency longitudinal modulus measured by Brillouin scattering is, however, fundamentally different from the low-frequency, tensile (Young's) modulus often-used in mechanobiology. Further research is thus needed to consolidate the mechanical measurements obtained by Brillouin microscopy with other work in the field.

Fig. R2. Experimental Brillouin shift image of a fibroblast cell on a 12kPa PAA gel measured by confocal Brillouin microscope at 532 nm. The color legend is in GHz.

Except of phototoxicity and high powers needed to register sufficient Brillouin spectra, Brillouin microscopy suffers many limitations. e.g. in this setup the sample thickness and optical transparency needs to be very tightly controlled. I encourage the authors to face these limitations in the introduction section and through the manuscript. Otherwise one could have a feeling that SBS can be directly implemented in clinics for mechanotyping of biopsy specimens or other urgent application. I also suggest expanding the reference list, for instance the group of Prof. Jochen Guck publish a lot on BM measurements of biological specimens (like spinal cord development).

We fully agree with the Reviewer's comment and note that we already refer to the limitations in terms of sample thickness and optical transparency in the discussion section of the original manuscript (line 291-293: "The limitations of pulsed-SBS are that it is presently restricted to relatively transparent (i.e., non-absorbing) samples thinner than ~100-200 μm that can be optically accessed from two opposing sides."). We realize that such a discussion is maybe better placed in the introduction section, and have thus moved this sentence accordingly.

Furthermore, we have also expanded the reference list following the suggestion of the Reviewer. While we had already cited work by the Guck lab on spinal cord growth and repair (Ref. 14 in the original version, Ref. 12 in the revised version), we have included further papers by this group in the revised Discussion section.

Changes to References:

45. Schlübler, R. et al. Correlative all-optical quantification of mass density and mechanics of sub-cellular compartments with fluorescence specificity. *Elife* 11, 1–23 (2022).
46. Bakhshandeh, S. et al. Optical quantification of intracellular mass density and cell mechanics in 3D mechanical confinement. *Soft Matter* 17, 853–862 (2021).

Decision Letter, first revision:

Dear Dr. Prevedel,

Thank you for submitting your revised manuscript "Pulsed stimulated Brillouin microscopy enables high-sensitivity mechanical imaging of live and fragile biological specimens" (NMETH-A51474A). It has now been seen by the original referees and their comments are below. The reviewers find that the paper has improved in revision, and therefore we'll be happy in principle to publish it in Nature Methods, pending minor revisions to satisfy to comply with our editorial and formatting guidelines.

TRANSPARENT PEER REVIEW

Nature Methods offers a transparent peer review option for new original research manuscripts submitted from 17th February 2021. We encourage increased transparency in peer review by publishing the reviewer comments, author rebuttal letters and editorial decision letters if the authors agree. Such peer review material is made available as a supplementary peer review file. Please state in the cover letter 'I wish to participate in transparent peer review' if you want to opt in, or 'I do not wish to participate in transparent peer review' if you don't. Failure to state your preference will result in delays in accepting your manuscript for publication.

ORCID

Thank you again for your interest in Nature Methods. Please do not hesitate to contact us if you have any questions. We will be in touch again soon.

Sincerely yours,
Allison

Allison Doerr, Ph.D.
Chief Editor
Nature Methods

On behalf of:

Nina Vogt, PhD
Senior Editor
Nature Methods

Reviewer #2 (Remarks to the Author):

The authors have satisfactorily addressed most of my concerns.

Reviewer #3 (Remarks to the Author):

The authors have satisfactorily addressed my comments and improved the manuscript. Now, I recommend the manuscript to be published in Nature Methods.

Author Rebuttal, first revision:

None required

Final Decision Letter: